# IV-BENCH: A BENCHMARK FOR IMAGE-GROUNDED VIDEO PERCEPTION AND REASONING IN MULTIMODAL LLMS

**David Ma**[1]*, **Yuanxing Zhang**[2]*, **Jincheng Ren**[1], **Jiawei Guo**[1], **Yifan Yao**[1], **Zhenlin Wei**[1]
**Zhenzhu Yang**[1], **Zhongyuan Peng**[1], **Boyu Feng**[1], **Jun Ma**[1], **Xiao Gu**[1], **Zhoufutu Wen**[1],
**King Zhu**[1] **Yancheng He**[1], **Meng Cao**[3], **Shiwen Ni**[4,5], **Jiaheng Liu**[1,6], **Wenhao Huang**[1],
**Ge Zhang**[1],†, **Xiaojie Jin**[1]†
[1]M-A-P    [2]Kuaishou Technology    [3]MBZUAI    [4]AIRI,SUAT    [5]SIAT,CAS    [6]NJU

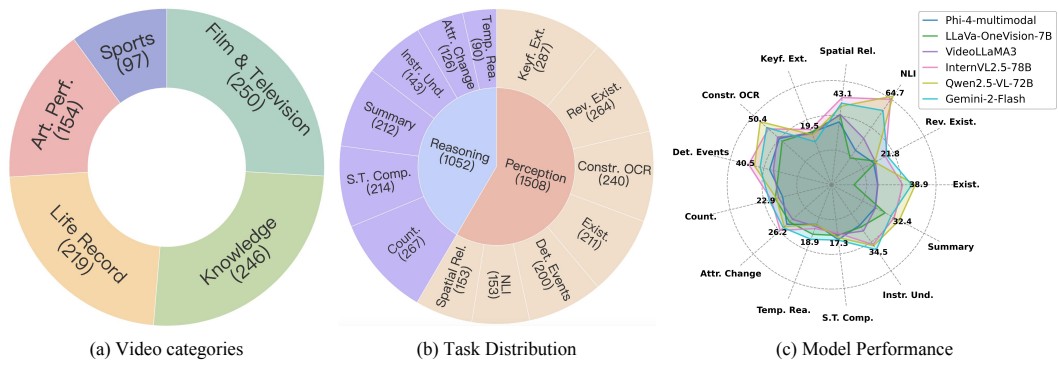

(a) Video categories      (b) Task Distribution      (c) Model Performance

Figure 1: (a) Video Categories. IV-Bench includes videos spanning five representative categories, ensuring diverse topical coverage. (b) Task distribution in IV-Bench. IV-Bench consists of a total of 13 tasks, which are categorized into two main types: 6 reasoning tasks and 7 perception tasks. (c) Model Performance on IV-Bench. All evaluated MLLMs exhibit limited performance on IV-Bench. Even on the best-performing task (Natural Language Inference), the highest achieved accuracy is merely 64.7%, with other tasks resulting in substantially lower scores.

## ABSTRACT

Current benchmarks for Multimodal Large Language Models (MLLMs) predominantly rely on text-only queries, overlooking the essential role of images as visual context for enhancing video comprehension and facilitating natural human-AI interaction. To bridge this gap, we introduce **IV-Bench**, the first comprehensive benchmark for evaluating MLLMs on *Image-Grounded Video Perception and Reasoning*. IV-Bench comprises 966 videos paired with 2,560 meticulously annotated image-text queries across 13 tasks (7 perception and 6 reasoning tasks) spanning 5 distinct categories. We extensively evaluate state-of-the-art MLLMs, including open-source models (e.g., InternVL2.5, Qwen2.5-VL) and closed-source models (e.g., GPT-4o, Gemini2.0 series), revealing substantial performance gaps, with the best-performing model achieving only 28.9% accuracy. Ablation studies demonstrate that incorporating images significantly enhances video understanding and highlight key model design factors influencing performance. Our findings provide valuable insights and guidance for future research. The code and dataset are available at `https://github.com/multimodal-art-projection/IV-Bench`.

---

*Equal contribution.
†Corresponding author.

# 1 INTRODUCTION

Building upon the remarkable success of Large Language Models (LLMs) across various AI tasks (Young et al., 2024; Zhang et al., 2024b), Multimodal Large Language Models (MLLMs) have recently demonstrated impressive capabilities in comprehending and integrating multimodal information from diverse sources such as text, images, and videos (Liu et al., 2023; Chen et al., 2024c; Guo et al., 2024). Consequently, numerous evaluation benchmarks (Yue et al., 2024; Zhang et al., 2024a; Hu et al., 2025; Cheng et al., 2025; Wu et al., 2024a;b; Zhu et al., 2024) have emerged to systematically assess their multimodal understanding, integration, and task-solving capabilities.

Existing video benchmarks predominantly focus on text-based queries, which fundamentally struggle to convey precise visual detail. Language is inherently too abstract for specifying fine-grained attributes (e.g., subtle design variations) or identifying specific instances (e.g., a person's face). This limitation is not just theoretical; it is underscored by the massive real-world adoption of visual search (Zhang et al., 2018). For instance, Pinterest's Lens was handling 600 million visual searches per month as early as 2018 (Zhai et al., 2017), and research shows 62% of young consumers desire visual search capabilities precisely because they excel where words fail (Dagan et al., 2023). A more detailed discussion is provided in Appendix A.5. An external image query overcomes this ambiguity by providing a concrete visual anchor, a capability critical in applications from e-commerce to media analysis. We define this fundamental challenge as image-grounded video perception and reasoning. Despite its proven importance and widespread applicability, this capability remains systematically overlooked in current evaluation frameworks.

To bridge this significant research gap, we propose **IV-Bench**, *the first benchmark specifically designed for evaluating MLLMs in image-grounded video perception and reasoning*. IV-Bench includes 966 diverse videos paired with 2,560 meticulously annotated image-text queries distributed across 13 tasks (7 perception and 6 reasoning) and 5 categories: Knowledge, Film & Television, Sports Competition, Artistic Performance, and Life Record (see Figure 1(a)). Critically, IV-Bench's use of externally sourced reference images ensures that the evaluation transcends a simple frame-matching task. This prevents information leakage and forces models to rely on generalized knowledge rather than perceptual similarity, offering a more rigorous and valid test of their real-world understanding.

Through extensive evaluations on 28 MLLMs—including 23 advanced open-source models like InternVL2.5 (Chen et al., 2024b), Qwen2.5-VL series (Bai et al., 2025), and 5 leading closed-source models such as GPT-4o (OpenAi, 2024) and Gemini-2.0 series (Team et al., 2024)—we uncover substantial performance deficiencies. A significant disparity is evident between model and human capabilities, with the best models achieving only 28.9% accuracy, in stark contrast to the 88.8% achieved by humans. These limitations become even more pronounced in complex reasoning tasks such as Temporal Reasoning. Our analysis provides key insights to guide future research. Ablation studies reveal that the ability to utilize visual context is critically dependent on model scale, with larger models showing significant benefits while smaller models often fail to do so. We further dissect the impact of visual token allocation, demonstrating that performance is generally more sensitive to increasing the number of video frames than to enhancing resolution. These findings, combined with our discovery that placing the image query after video frames yields the best results, provide crucial insights for advancing the capabilities of MLLMs on image-grounded video tasks.

In summary, our work has three-fold contributions:

1. We present IV-Bench, the first comprehensive benchmark for image-grounded video perception and reasoning in MLLMs. IV-Bench comprises 966 videos paired with 2,560 meticulously annotated image-text queries, where the images, collected from external sources rather than extracted from the videos themselves, provide the essential context required to accurately answer the queries. The dataset spans 5 major categories and covers 7 perception and 6 reasoning tasks, ensuring substantial diversity across various scenarios and task types. Moreover, two round quality control—one ensuring clarity, accuracy, and category labeling, and another confirming that both image and video are required to answer correctly, ensuring the high quality of IV-Bench.

2. We evaluate 28 state-of-the-art MLLMs, including the latest closed-source models (e.g., GPT-4o and Gemini-2.0-Pro) and open-source models (e.g., InternVL2.5 and Qwen2.5-VL series). Our experiments reveal that current models perform poorly on IV-Bench, with the best model

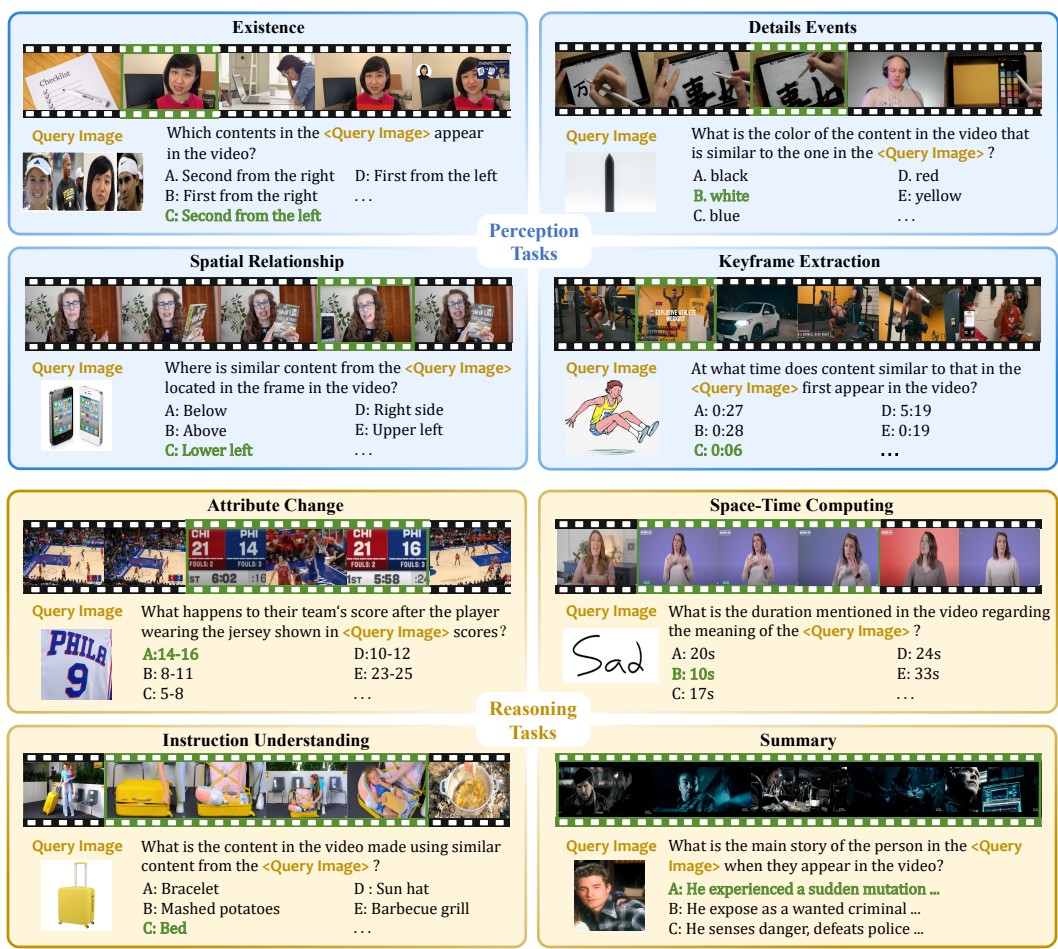

Figure 2: Representative examples from IV-Bench. Each sample consists of a video paired with an image-text query, comprising a query image and corresponding query text. The correct answer is marked in green, with relevant video frames also highlighted in green.

achieving only 28.9% overall accuracy—clearly indicating an big research gap for enhancing the capablilities of MLLMs in this direction.

3. Our analysis provides key insights to guide future research. Ablation studies indicate that increasing frame number and video resolution positively affect performance. Moreover, larger models significantly benefit from image contexts presented after the video, while smaller models show minimal improvements.

## 2 IV-BENCH

### 2.1 OVERVIEW

IV-Bench is designed to evaluate image-grounded video perception and reasoning, aiming to assess the capabilities of MLLMs in utilizing external visual cues for localization, reasoning, and comprehension of video content. IV-Bench consists of 966 diverse videos paired with 2,560 image-text queries, with each image providing indispensable contextual cues necessary for correctly answering the queries. The dataset spans 13 distinct tasks. Key features of IV-Bench include:

**Image-Text Queries.** For each video, we design multiple image-text queries. Each query includes an externally sourced image—not extracted from the video itself—and an associated textual question. These externally sourced images guarantee greater visual diversity and better simulate

real-world usage scenarios, providing critical contextual cues necessary to accurately answer the queries.

**Diverse Video Categories.** The dataset covers a wide array of categories including Knowledge, Film & Television, Sports Competitions, Artistic Performances, and Life Records, ensuring extensive content diversity for various research purposes. Each video, with a minimum duration of five minutes, provides sufficient depth for comprehensive analysis.

**Diverse Evaluation Tasks.** As illustrated in Figure 1(b), IV-Bench offers 13 distinct evaluation tasks grouped into perception and reasoning categories. These tasks comprehensively assess the capability of MLLMs to perform image-grounded video perception and reasoning, spanning a diverse set of perceptual and reasoning skills.

## 2.2 TASK DEFINITIONS

IV-Bench comprises 7 perception tasks and 6 reasoning tasks, with representative examples of selected tasks shown in Figure 2. Additional examples of the remaining tasks are provided in Figure 5. These tasks address various aspects of image-grounded video comprehension, spanning from basic perception to complex reasoning. Detailed descriptions of each task are presented below.

**Perception Tasks:** These tasks evaluate the model's capability to directly extract and interpret fundamental visual information from the video. They primarily focus on recognizing objects, people, scenes, and spatial relationships by leveraging contextual cues from the reference image. In essence, perception tasks assess the model's ability to accurately 'see' and identify content within the video. *1) Existence:* Identify which objects or people in the reference image appear in the video. *2) Reverse Existence:* Identify objects or people present in the image but absent in the video. *3) Natural Language Inference:* Determine which scenes in the video are similar to a specific scene in the image. *4) Spatial Relationship:* Identify absolute or relative spatial relationships among objects or people in the video, grounded by the reference image. *5) Keyframe Extraction:* Identify precise timestamps or segments within the video where objects or people depicted in the image appear. *6) Constrained OCR:* Recognize text-based content in the video, constrained by conditions explicitly defined by the reference image, such as spatial alignment, temporal correspondence, and semantic relevance. *7) Detailed Events:* Identify specific events or actions within the video directly related to content depicted in the reference image.

**Reasoning Tasks:** These tasks require models to engage in higher-order cognitive functions by integrating visual cues with contextual and temporal information. They assess the model's capacity to analyze, synthesize, and infer meaningful conclusions beyond simple visual recognition. *1) Counting:* Count occurrences of a person, object, or action depicted in the video, grounded by the reference image. *2) Space-Time Computing:* Calculate event durations or distances between objects/people in the video, using the image as contextual guidance. *3) Summary:* Generate a brief description summarizing a person, object, or event depicted in the video, informed by the reference image. *4) Instruction Understanding:* Understand the functionality or creation process of objects depicted in the video, guided by the reference image. *5) Attribute Change:* Detect changes in attributes (e.g., clothing, size, color) of objects or people throughout the video, referenced by the image. *6) Temporal Reasoning:* Infer precise start and end timestamps of target events using temporal cues and world knowledge, as introduced in (Huang et al., 2024).

## 2.3 ANNOTATION AND QUALITY CONTROL

The high quality of IV-Bench is ensured by a rigorous, two-stage quality control process. Initially, annotators select external images from non-video sources and formulate questions with multiple plausible distractors. The first quality control round verifies fundamental clarity and accuracy. The second round is more stringent: we eliminate samples solvable without the image and, critically, ensure each question includes at least two 'effective' distractors. These distractors are designed to be plausible yet incorrect, but would become correct answers if paired with a different image, thus confirming the necessity of the provided image for accurate reasoning.

### 2.3.1 ANNOTATION PROCESS

**Video Collection**: A total of 966 videos, each exceeding five minutes in length, are carefully selected to ensure broad topical coverage. These videos span five distinct categories: Knowledge, Film & Television, Sports, Artistic Performances, and Life Records, providing diverse content suitable for various research applications.

**Task Assignment**: Annotators first watch each video in its entirety before performing image and text annotations. They then assign an appropriate task type from 13 predefined categories. This initial task assignment guides subsequent annotations, ensuring alignment with task-specific requirements.

**Image and Question Annotation**: After task assignment, annotators manually retrieve a relevant external image from online sources, explicitly ensuring it is not extracted from the video itself. The selected image must be closely related to specific keywords, individuals, or themes present in the video. Annotators then formulate a text question leveraging both the video content and contextual cues provided by the image. The diversity of these external images is presented in Appendix A.6, and their necessity for the task is justified in Appendix A.7.

**Answer and Distractor Design**: Annotators craft the correct answer by carefully analyzing multimodal information from both the video and the image. Additionally, they generate up to nine plausible yet incorrect distractors to increase the question's difficulty while ensuring contextual relevance. For certain questions, fewer distractors may be provided depending on content constraints.

### 2.3.2 QUALITY CONTROL

Ensuring the quality and consistency of the data is crucial for creating a reliable dataset. Our quality control process is conducted in two main rounds (see Appendix A.2 for more detail of quality control):

**First Round Quality Control**: The first round quality check focuses on the structure and content standardization of evaluation questions. It mainly checks whether the query and options are clearly described, whether the answer is correct, and whether the distractors can effectively mislead test-takers. So we verify the clarity, precision, and unambiguity of each question. We also confirm that the correct answers and distractors are both plausible and contextually relevant, ensuring each query can be accurately answered based on the provided video and image. Furthermore, we check task categorization accuracy, correcting any misclassifications to maintain consistency across all 13 predefined tasks.

**Second Round Quality Control**: Since some questions can be answered using only common sense or video content—we conduct a second round of quality check. During this phase, any query that can be resolved without the reference image or video is simply removed, and any text query that inadvertently reveals visual content is rewritten to eliminate leakage. We also pinpoint ineffective distractors—those easily dismissed using video alone—and manually introduce at least two effective distractors per question; These distractors are crafted so that, although incorrect for the current image, they would serve as the correct answers to alternative questions sharing the same text query but paired with a different image—thereby ensuring that the image is necessary for each sample in IV-Bench.

## 3 COMPARISON WITH OTHER VIDEO BENCHMARKS

As shown in Table 1, existing approaches can be broadly categorized into two groups: benchmarks with text-only queries and those with combined image-text queries. Benchmarks with text-only queries primarily emphasize evaluating video understanding guided solely by textual instructions. Existing benchmarks—from short-video suites like MMBench-Video (Fang et al., 2025) and Video-Bench (Ning et al., 2023) to long-range collections such as LongVideoBench (Wu et al., 2025a), MLVU (Zhou et al., 2024), and Video-MME (Hu et al., 2025)—span varied durations and modalities but none employ image-text queries, limiting their ability to evaluate image-grounded video perception and reasoning.

Although V2P-Bench (Zhao et al., 2025) and VideoRefer-Bench (Yuan et al., 2025) both employ image–text queries, all images are sourced from within the videos themselves; nonetheless, relying

Table 1: A comparison of representative video benchmarks. ImgSrc, ImgNec, and VidNec are abbreviations for Image Source, Image Necessity, and Video Necessity, respectively.

| Query Modality | Benchmark | #Videos | Duration. (s) | #Tasks | #QA Pairs | Anno. | # Avg. Opt. | ImgSrc | ImgNec | VidNec |
|---|---|---|---|---|---|---|---|---|---|---|
| | MMBench-Video | 609 | 165 | 26 | 1,998 | M | - | - | - | ✓ |
| | Video-Bench | 5,917 | 56 | 10 | 17,036 | M & A | 4 | - | - | ✓ |
| | EgoSchema | 5,063 | 180 | - | 5,063 | M & A | 5 | - | - | ✓ |
| | AutoEval-Video | 327 | 14.6 | - | 327 | A | - | - | - | ✓ |
| Text | TempCompass | 410 | 11.4 | - | 7,540 | M & A | - | - | - | ✓ |
| | MVBench | 3,641 | 16 | 20 | 4,000 | M | 4 | - | - | ✓ |
| | LongVideoBench | 3,763 | 473 | 17 | 6,678 | M | 4 | - | - | ✓ |
| | MLVU | 1,730 | 930 | 9 | 3,102 | M & A | 4 or 6 | - | - | ✓ |
| | Video-MME | 900 | 1,017.9 | 12 | 2,700 | M | 4 | - | - | ✓ |
| Text+Image | VideoRefer-Bench[Q] | 198 | - | 5 | 1000 | M | 4 | In-Video | ✓ | ✓ |
| Text+Image | V2P-Bench | 980 | 1140 | 12 | 1172 | M | 4 | In-Video | ✓ | ✓ |
| Text-only/Text+Image | Video-MMMU | 300 | 506.2 | 6 | 900 | M | 10 | Out-of-Video | ✗ | ✗ |
| Text+Image | IV-Bench | 966 | 537 | 13 | 2,560 | M | 9 | Out-of-Video | ✓ | ✓ |

solely on frame-extracted visuals constrains content diversity and real-world applicability. Video-MMMU (Hu et al., 2025) is the only video-QA benchmark employing external images, but there the video serves merely as an auxiliary to boost image-only or text-only performance and isn't required to answer the questions. In contrast, IV-Bench is the first manually annotated image-grounded video perception and reasoning benchmark: each query pairs an external image with its corresponding video—both of which are indispensable for deriving the correct answer. Refer to the Appendix for illustrative examples that compare IV-Bench and VideoMMMU.

## 4 EXPERIMENTS

This section evaluates several representative MLLMs on IV-Bench. We begin by outlining the models and experimental setup, then analyze the performance of the 28 MLLMs in IV-Bench, and finally examine how inference patterns and how the number of visual tokens influences results.

### 4.1 SETTINGS

We evaluate 5 commercial models: Doubao-1.5-vision-pro (Doubao Team, 2025), Claude-3.7-sonnet (Anthropic, 2025), GPT-4o (OpenAi, 2024), Gemini 2 Flash (Team et al., 2024), and Gemini 2 Pro (Team et al., 2024). For open-source models, we select 23 representative MLLMs, including the Qwen2.5-VL series, InternVL2.5-VL series, and VideoLLaMA3 (Zhang et al., 2025a). We employ a uniform sampling strategy to process video frames. For all models, we uniformly set the frame number to 32. The default evaluation input format is 'video frames + image + question' with prompts, indicating that video frames are provided first, followed by the image. Since test samples in IV-Bench are multiple-choice questions, accuracy is computed by directly matching the model's output to the correct answer.

### 4.2 MAIN RESULTS

The performance across the 13 IV-Bench tasks—7 perception tasks and 6 reasoning tasks—is presented in Table 2. We report individual task performances along with average results for perception tasks (P-Avg) and reasoning tasks (R-Avg). From the results, we derive two primary conclusions:

**Image-Grounded video perception and reasoning is Challenging**: In stark contrast to human performance, which achieves an average accuracy of 88.8% (see Appendix A.8), effectively performing image-grounded video perception and reasoning continues to pose significant challenges for MLLMs. For instance, among models under 10B parameters, the top-performing Qwen2.5-VL-7B achieves merely 18.5% accuracy, whereas random guessing would yield 11.11%. Even larger models such as InternVL2.5-78B and Qwen2.5-VL-72B achieve just 28.6% and 28.9%, respectively. Moreover, the best-performing commercial model, Gemini-2.0-Pro, reaches only 27.7%. This vast gap to human-level performance underscores the significant untapped potential in leveraging image contexts to improve video comprehension. Overall, perception tasks generally prove easier for models than reasoning tasks, though both remain far from human proficiency. For example, InternVL2.5-78B achieves a perception task average (P-Avg) of 33.4%, compared to a reasoning task average (R-Avg) of only 21.9%. Even Gemini-2-Pro, which performs relatively better on reasoning tasks, manages only 24.9%. Temporal reasoning remains especially challenging, with the

Table 2: The performance of MLLMs on IV-Bench across 13 tasks, comprising 7 perception tasks and 6 reasoning tasks. For each task, the best performance is indicated in bold, and the second-best performance is indicated with underlining. Note that '**P-Avg**' and '**R-Avg**' denote the average results on perception and reasoning tasks, respectively.

| Models | Overall | Perception | | | | | | | | Reasoning | | | | | | |
|---|---|---|---|---|---|---|---|---|---|---|---|---|---|---|---|---|
| | | Exist. | RE | NLI | SR | KE | CO | DE | P-Avg | Cnt. | AC | TR | STC | IU | Sum. | R-Avg |
| Human | 88.8 | 91.0 | 89.0 | 92.2 | 93.5 | 90.3 | 95.4 | 89.0 | 91.5 | 74.5 | 81.9 | 87.9 | 98.1 | 88.1 | 91.1 | 86.9 |
| *Open Source MLLMs(< 10B)* | | | | | | | | | | | | | | | | |
| Llama-vid (Li et al., 2024d) | 10.5 | 13.3 | 10.6 | 6.5 | 7.2 | 9.8 | 13.3 | 16.0 | 11.2 | 7.1 | 8.7 | 2.2 | 12.2 | 11.2 | 12.3 | 9.5 |
| LLaVA-Mini (Zhang et al., 2025b) | 12.5 | 7.1 | 12.1 | 9.8 | 15.0 | 10.8 | 12.9 | 18.0 | 12.1 | 12.7 | 13.5 | 13.3 | 14.0 | 14.7 | 11.3 | 13.1 |
| MAmmoTH-VL-8B (Guo et al., 2024) | 13.3 | 3.3 | 8.7 | 3.9 | 29.4 | 10.5 | 19.6 | 14.5 | 12.4 | 12.7 | 16.7 | 7.8 | 13.1 | 14.0 | 20.3 | 14.5 |
| Longva (Zhang et al., 2024d) | 14.4 | 5.2 | 20.1 | 4.6 | 20.3 | 13.9 | 27.5 | 19.5 | 16.4 | 7.5 | 11.9 | 14.4 | 15.0 | 10.5 | 13.2 | 11.7 |
| NVILA (Liu et al., 2024b) | 14.4 | 2.4 | 17.4 | 2.6 | 22.2 | 13.2 | 24.6 | 14.0 | 14.2 | 12.7 | 18.3 | 18.9 | 17.3 | 14.7 | 10.8 | 14.7 |
| Longvu (Shen et al., 2024) | 14.8 | 3.8 | 17.1 | 11.1 | 20.9 | 11.5 | 19.6 | 19.5 | 14.7 | 18.0 | 16.7 | 8.9 | 15.9 | 16.8 | 10.9 | 15.0 |
| Phi-3.5-vision (Microsoft, 2024) | 15.2 | 5.2 | 12.9 | 11.1 | 23.5 | 10.8 | 25.0 | 22.5 | 15.5 | 16.1 | 15.9 | 11.1 | 13.6 | 14.7 | 15.6 | 14.8 |
| Phi-4-multimodal (Abdin et al., 2024) | 16.6 | 11.8 | 12.5 | 9.8 | 22.4 | 17.5 | 27.2 | 22.6 | 17.8 | 14.1 | 19.8 | 10.0 | 17.0 | 13.5 | 13.5 | 14.8 |
| LLaVA-OneVision-7B (Li et al., 2024a) | 16.3 | 2.8 | 14.4 | 5.9 | 27.5 | 17.8 | 24.2 | 16.5 | 15.7 | 18.7 | 17.5 | 15.6 | 14.5 | 14.7 | 20.3 | 17.2 |
| InternVL2-8B (OpenGVLab, n.d.) | 16.8 | 7.1 | 10.6 | 19.0 | 25.5 | 12.2 | 27.9 | 22.5 | 17.1 | 18.7 | 20.6 | 6.7 | 14.5 | 14.7 | 17.9 | 16.3 |
| VAMBA (Ren et al., 2025) | 16.9 | 13.7 | 11.7 | 13.7 | 26.1 | 12.9 | 27.9 | 22.0 | 17.8 | 15.7 | 21.4 | 12.2 | 13.6 | 13.3 | 16.5 | 15.5 |
| Minicpm-v (Yao et al., 2024) | 17.2 | 7.1 | 15.5 | 9.2 | 30.1 | 14.3 | 32.5 | 22.5 | 18.6 | 15.7 | 15.9 | 12.2 | 15.0 | 13.3 | 17.5 | 15.3 |
| Minicpm-o (Yao et al., 2024) | 17.1 | 8.1 | 14.8 | 8.5 | 28.8 | 14.6 | 31.2 | 23.5 | 18.4 | 13.1 | 13.5 | 12.2 | 17.8 | 16.8 | 16.5 | 15.2 |
| VideoLLaMA3 (Zhang et al., 2025a) | 17.4 | 11.9 | 13.6 | 17.7 | 28.1 | 16.4 | 28.9 | 20.0 | 19.0 | 16.5 | 15.1 | 11.1 | 14.0 | 17.5 | 13.7 | 14.9 |
| InternVL2.5-8B (Chen et al., 2024b) | 17.4 | 5.4 | 15.7 | 16.4 | 28.3 | 10.1 | 31.4 | 25.2 | 17.8 | 15.7 | 22.1 | 8.2 | 14.1 | 16.3 | 20.6 | 16.5 |
| Qwen2.5-VL-7B (Bai et al., 2025) | 18.5 | 8.1 | 13.6 | 21.6 | 26.8 | 15.7 | 31.7 | 18.5 | 18.9 | 16.5 | 23.0 | 6.7 | 15.4 | 17.5 | 24.1 | 17.9 |
| *Open Source MLLMs(> 10B)* | | | | | | | | | | | | | | | | |
| Aria (Li et al., 2024b) | 17.4 | 9.5 | 9.5 | 11.8 | 24.8 | 17.4 | 36.7 | 20.5 | 18.6 | 16.9 | 16.7 | 15.6 | 14.0 | 18.9 | 13.7 | 15.8 |
| InternVL2.5-26B | 20.6 | 12.8 | 14.4 | 33.3 | 36.6 | 13.6 | 32.5 | 24.5 | 22.4 | 15.0 | 23.0 | 14.4 | 14.0 | 25.2 | 19.8 | 18.1 |
| LLaVA-OneVision-72B | 22.3 | 15.6 | 18.9 | 40.5 | 28.8 | 20.1 | 32.9 | 27.0 | 25.2 | 17.6 | 17.5 | 18.9 | 15.3 | 21.1 | 20.5 | 18.3 |
| Qwen2.5-VL-32B | 23.0 | 13.3 | 22.8 | 28.8 | 31.6 | 13.8 | 41.7 | 27.1 | 24.9 | 20.5 | 23.0 | 12.2 | 14.7 | 25.4 | 23.6 | 20.2 |
| InternVL2.5-38B | 26.6 | 24.6 | 19.3 | 56.9 | 34.0 | 19.5 | 38.8 | 33.5 | 30.4 | 20.2 | 22.2 | 6.7 | 22.0 | 27.3 | 22.6 | 21.1 |
| InternVL2.5-78B | 28.6 | 28.0 | 20.5 | 60.1 | 43.1 | 19.5 | 39.6 | 40.5 | 33.4 | 21.4 | 26.2 | 12.2 | 13.1 | 29.4 | 27.8 | 21.9 |
| Qwen2.5-VL-72B | 28.9 | 38.9 | 13.3 | 64.7 | 34.9 | 16.0 | 50.4 | 35.0 | 33.7 | 17.9 | 22.2 | 10.1 | 15.7 | 30.5 | 32.4 | 21.9 |
| *Closed Source MLLMs* | | | | | | | | | | | | | | | | |
| Doubao-1.5-vison-pro(Doubao Team, 2025) | 19.2 | 20.9 | 12.5 | 19.0 | 26.1 | 14.7 | 31.7 | 26.5 | 21.0 | 12.8 | 23.0 | 17.8 | 14.4 | 15.1 | 20.1 | 16.5 |
| Claude-3.7-sonnet (Anthropic, 2025) | 21.7 | 33.0 | 17.6 | 52.7 | 21.1 | 12.5 | 29.3 | 30.5 | 26.2 | 11.2 | 12.7 | 7.9 | 15.2 | 21.1 | 20.4 | 15.1 |
| GPT-4o (OpenAi, 2024) | 26.1 | 31.3 | 20.5 | 47.1 | 32.7 | 13.8 | 35.0 | 37.5 | 29.3 | 13.1 | 18.3 | 13.3 | 19.2 | 33.6 | 31.6 | 21.5 |
| Gemini-2-Flash (Team et al., 2024) | 27.4 | 35.6 | 21.8 | 45.4 | 37.7 | 11.8 | 41.7 | 30.1 | 30.2 | 22.9 | 23.2 | 18.9 | 17.3 | 34.5 | 25.0 | 23.4 |
| Gemini-2-Pro (Team et al., 2024) | 27.7 | 26.4 | 23.8 | 35.5 | 39.1 | 16.4 | 43.8 | 30.1 | 29.6 | 22.6 | 21.6 | 15.6 | 19.7 | 39.6 | 29.5 | 24.9 |

top-performing model achieving only 16.7% accuracy. Conversely, NLI and existence tasks appear relatively easier, with Qwen2.5-VL-72B scoring 38.9% and 64.7%, respectively, yet these scores still highlight the profound difficulty for models compared to humans.

**Moderate Gains with Larger Models**: Increasing model scale results in modest performance improvements across various tasks. For example, in the InternVL2.5 series, the 8B model achieves 17.4% accuracy, while the larger 26B, 38B, and 78B variants reach 20.6%, 26.6%, and 28.6%, respectively. Similarly, for the Qwen2.5-VL series, scaling from the 7B model (18.5%) to the 72B model (28.9%) results in noticeable performance gains. This improvement is particularly pronounced in few perception-based tasks. For example, Qwen2.5-VL-72B demonstrates a substantial 30.8% improvement over Qwen2.5-VL-7B on existence tasks and a notable 43.1% gain on NLI tasks. However, the benefits of scaling are notably smaller for reasoning-intensive tasks. For example, on counting tasks, Qwen2.5-VL-72B surpasses Qwen2.5-VL-7B by merely 1.4%, while InternVL2.5-78B exhibits just a 5.7% improvement over InternVL2.5-8B. The limited benefits observed on reasoning-intensive tasks may be attributed to the fact that increasing model size tends to enhance memorization and shallow pattern recognition more significantly than improves reasoning ability. More detail can be seen in Appendix A.10

The experimental results confirm that image-grounded video perception and reasoning continue to pose significant challenges for MLLMs, especially for reasoning-intensive tasks. Furthermore, increasing model size yields only moderate improvements, suggesting that merely scaling models is insufficient to fully overcome these challenges. Future research should prioritize developing specialized mechanisms for video reasoning, such as enhanced temporal modeling techniques.

## 4.3 THE IMPACT OF INFERENCE PATTERN

In this section, we comparatively analyze performance across three inference patterns: image-first, video-first (both under image-text query settings), and text-only queries (without image input). We evaluate both the MiniCPM-v/o and InternVL2.5 model series, with results presented in Figure 3(a).

**Smaller models are less proficient in image-grounded video perception and reasoning.** MiniCPM-v/o achieve its highest scores under the text-only setting, whereas incorporating im-

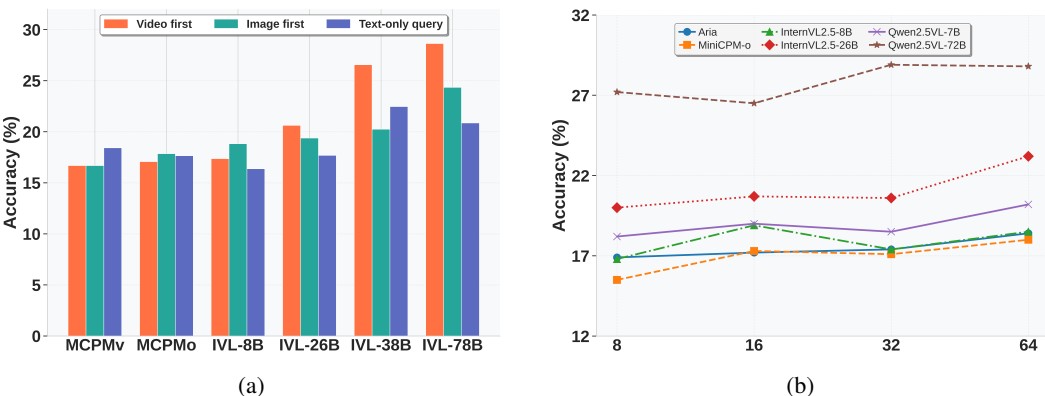

Figure 3: Comparison of model performance: (a) across different inference patterns and (b) with varying numbers of frames. MCPMv/o represent MiniCPMv/o, IVL is the abbreviation of InternVL.

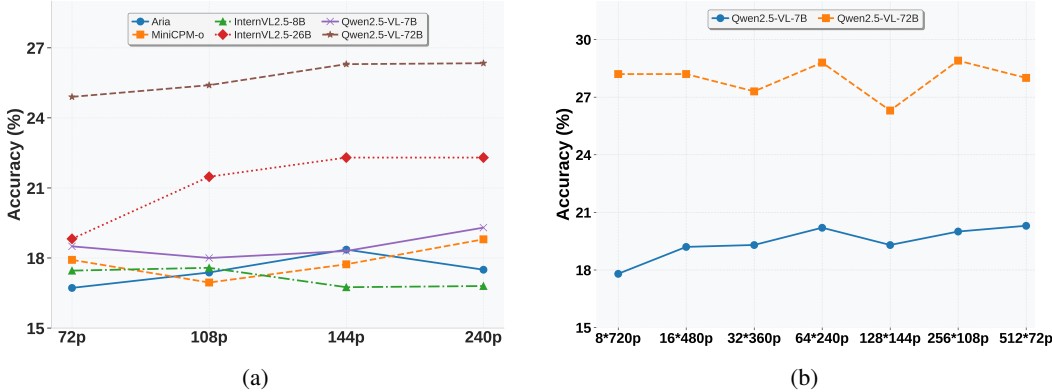

Figure 4: Comparison of model performance: (a) across different video resolutions and (b) across various frame-resolution combinations.

ages—whether before or after the video—leads to a drop in performance. This suggests that a lack of sufficient image-grounded video training data, combined with limited model capacity, renders visual context unhelpful or even detrimental for smaller architectures. InternVL2.5-8B likewise shows only marginal gains when images are added, reinforcing this trend.

**For larger models, placing the image after video frames results in superior performance compared to positioning it beforehand.** Across InternVL2.5-VL variants of 26B, 38B, and 78B parameters, we observe steadily improving results when the image is appended after the sequence of video frames. We hypothesize that images presented early in the token stream are more easily 'forgotten' amid subsequent video tokens, leading to under-utilization of crucial visual information—a phenomenon similarly reported in (Ma et al., 2024). More analysis can be seen in Appendix A.12

## 4.4 ANALYSIS OF THE NUMBER OF VISUAL TOKENS

This section addresses two key questions regarding the number of visual tokens supplied to multimodal vision–language models:

**Scaling Effect.** How does model performance change as we increase the total number of visual tokens—by varying either frame number or resolution?

**Token Allocation.** When the total number of visual tokens is held constant, does allocating tokens to more frames or to higher resolution produce greater gains?

### 4.4.1 Isolated Scaling of Frame number and Resolution

In this subsection, we analyze how allocating additional visual tokens—both temporally (by increasing frame number) and spatially (by raising resolution)—impacts model performance. We conduct experiments on six top-performing models—Aria, MiniCPM-O, InternVL-8B/26B, and Qwen2.5-VL-7B/72B—under two settings. We find:

**Model performance consistently improves as frame number increases.** We vary frame number $\in \{8, 16, 32, 64\}$ at a fixed resolution (results in Figure 3(b)). Across all six models, increasing the number of frames steadily boosts accuracy, demonstrating that additional temporal visual tokens enhance image-grounded video perception and reasoning.

**Performance consistently improves as resolution increases—most markedly at low resolutions, though gains diminish at higher resolutions.** We fix frame number at 32 and vary resolution $\in \{72p, 108p, 144p, 240p\}$ (results in Figure 4(a)). When varying spatial detail, performance improves most between 72p and 144p, but further upgrades to 240p yield marginal gains, indicating diminishing returns beyond a mid-range resolution.

### 4.4.2 Token Allocation: Frames vs. Resolution

To disentangle temporal and spatial contributions under a fixed budget of visual tokens, we evaluate seven frame–resolution pairs—(8, 720p), (16, 480p), (32, 360p), (64, 240p), (128, 144p), (256, 108p), and (512, 72p)—chosen to yield roughly equivalent token number. Figure 4(b) reveals contrasting behaviors between model scales. For Qwen2.5-VL-7B, performance rises primarily with frame number, while resolution plays a secondary role. In contrast, Qwen2.5-VL-72B exhibits near-constant performance across all combinations, indicating its capacity to flexibly trade temporal for spatial information. These findings suggest that **smaller models rely more on temporal cues to compensate for limited spatial encoding, whereas larger models can extract complementary signals from both dimensions interchangeably.**

## 5 Related Work

### 5.1 Multimodal Large Language Models

Multimodal Large Language Models (MLLMs) fuse LLM backbones with visual encoders via visual instruction tuning for richer multimodal understanding (Liu et al., 2023; 2024a; Zhu et al., 2023; Ma et al., 2024; Zhang et al., 2024c; Bai et al., 2025). In video, they leverage instruction datasets and projectors (Cheng et al., 2024; Li et al., 2023)—e.g., Video-LLaMA's ViT+Q-Former (Zhang et al., 2023) and LLaMA-Vid's frame compression (Li et al., 2024d)—and can handle diverse tasks from captioning to temporal localization. However, despite training on massive image/video corpora (e.g., Qwen2.5-VL), they still lack targeted examples that ground video queries in external image contexts, limiting their ability to resolve cross-source visual references—motivating IV-Bench.

### 5.2 Video Understanding Benchmarks

Recent benchmarks target temporal perception (Wu et al., 2024), action understanding (Wang et al., 2023; Wu et al., 2024), and reasoning (Xiao et al., 2021), exemplified by MVBench for short-video QA (Li et al., 2024c), LongVideoBench's hour-long referring reasoning (Wu et al., 2025a), Video-MME's 11 s–1 h multimodal analysis (Fu et al., 2024), and V2P-Bench's 980 videos with 1,172 visual-prompt QA pairs (Zhao et al., 2025). However, these efforts focus only on intra-video signals and overlook the challenge of grounding video understanding in external image contexts. None, however, assess image-grounded video perception and reasoning. To fill this gap, we introduce IV-Bench—the first benchmark dedicated to evaluating image-grounded video perception and reasoning using images sourced externally from the videos themselves.

## 6 Conclusion

In this work, we introduce IV-Bench, the first benchmark explicitly designed to evaluate models on image-grounded video perception and reasoning tasks. Our extensive evaluation of both open-source

and closed-source multimodal large language models reveals significant limitations, particularly in effectively leveraging image contexts for accurate video comprehension, notably in temporally sensitive perception and reasoning tasks. We observe that smaller models demonstrate a limited capability of image grounded video perception and reasoning. Furthermore, our analysis indicates that increasing the number of video frames generally has a more pronounced impact on performance compared to enhancing video resolution. We hope IV-Bench will inspire future research to substantially advance the capabilities of multimodal large language models in image-grounded video perception and reasoning.

## 7   ETHICS STATEMENT

We have undertaken a thorough review of the ethical considerations related to the creation and deployment of the IV-Bench benchmark. Our ethical framework addresses three primary areas: data sourcing, potential biases, and human labor.

**Data Sourcing.**   The videos and external images comprising IV-Bench are sourced from publicly available online platforms. We have taken deliberate steps to filter this content, ensuring its suitability for a general research audience and excluding any material that is explicitly copyrighted, sensitive, or private. The use of this data is intended strictly for academic and non-commercial research purposes, in alignment with fair use principles.

**Potential Biases.**   We acknowledge that any dataset curated from web-based sources may inherit biases present in the original content. IV-Bench may reflect cultural, demographic, or subject-matter skews inherent to its sources. We transparently disclose this limitation to alert downstream users to the potential fairness risks. Models that perform well on IV-Bench may not generalize equitably across all cultural contexts or populations, and we encourage researchers to consider these factors when interpreting results.

**Human Labor.**   The development of IV-Bench involved human participation in two key stages: the classification of external images and the human performance evaluation. All annotators involved are experts with extensive experience in evaluating Multimodal Large Language Models (MLLMs) and complex visual reasoning tasks. They were fully informed about the scope and purpose of their tasks and provided their consent to be involved. All participants were compensated fairly for their expertise and time, in accordance with ethical standards for academic research and to respectfully acknowledge their valuable contributions.

## 8   REPRODUCIBILITY STATEMENT

We are fully committed to ensuring the reproducibility of our research. To facilitate the verification of our findings and to encourage future work, we make the IV-Bench dataset and our code publicly available at `https://github.com/multimodal-art-projection/IV-Bench`.

In the main body of our paper, particularly in the experimental setup section, we provide a detailed description of our evaluation protocol. This includes key hyperparameters and settings required for faithful replication, such as the number of sampled video frames (32), the input resolution (e.g., 480p), the specific input sequence used ($video \rightarrow image \rightarrow text$), and details of the models evaluated. We are confident that the public release of our benchmark and the comprehensive documentation of our methodology will allow the research community to easily reproduce our results and build upon our contributions.

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

# A APPENDIX

## A.1 ANNOTATION TUTORIAL

### A.1.1 QUESTION TYPE

🔧 Question Type Details

Representative examples of all 13 task categories in IV-Bench are shown in Figure 2 and Figure 5.

**Summary**

- These questions aim to test the model's ability, using the provided image, to understand the main narrative thread of the video and summarize the main plot or character stories from a global perspective. The model needs to leverage clues or context from the image, go beyond understanding single frames or segments, grasp the core content of the video, and comprehend and extract a plot summary.

**Spatial Relationship**

- These questions aim to test the model's ability, using the provided image, to understand the spatial positions of objects and their relationships within the video scene. The model needs to identify specific objects or areas based on the image, locate them in the video scene, and describe the surrounding spatial layout and relationships with other objects.

**Existence**

- These questions aim to test the model's ability, using the provided image, to identify a specific object within the image and search within the video content to determine if that object appears or is used in the video.

**Reverse Existence**

- These questions aim to test the model's ability, using the provided image (often showing a set), to perform a comparative analysis of the set and identify missing elements within the video content. The model needs to identify all items in the image set, compare them against the video content, and identify those items that do not appear in the video.

**Natural Language Inference**

- These questions aim to test the model's ability, using the provided image, to perform consistency reasoning between the visual content of the image and the video. The model needs to understand the image's visual information and the video's content to determine if the image is semantically consistent with the video content.

**Detailed Events Questions**

- These questions aim to test the model's ability, using the provided image, to identify a specific object or scene in the image, locate related events within the video, and extract specific detail information (e.g., price, time, location) from those events.

**Instruction Understanding**

- These questions aim to test the model's ability, using the provided image, to understand the attributes of the object shown in the image and perform an associative analysis connecting them with explanatory content in the video (such as introductions or descriptions). Based on the video content, the model needs to understand the reason, definition, function, impact, or creation process related to specific attributes of the object shown in the image.

**Keyframe Extraction**

- These questions aim to test the model's ability to integrate textual understanding with visual analysis of the provided image to identify a specific object or state. The model must then locate the corresponding keyframe(s) or segment within the video timeline, as depicted in the image, demonstrating comprehension of the 'keyframe' concept and the ability to correlate visual cues with temporal positioning.

**Counting**

- These questions aim to test the model's ability, using the provided image, to identify the specific object category designated in the image and count all instances of that object within the video scene(s). The model needs to accurately identify and differentiate between individual instances and report the total count.

**Space-Time Computing**

- These questions aim to test the model's ability, using the provided image (which might include scale information, a map, or specific reference objects), to understand spatial scales and track the movement trajectory of objects/people within the video, ultimately calculating the actual distance traveled. The model needs to utilize the scale or reference points from the image to analyze the motion depicted in the video.

**Space-Time Computing**

- These questions aim to test the model's ability, using the provided image (which might reference specific time points, events, or individuals), to understand and analyze temporal information and sequential events within the video. The model needs to use the clues from the image to perform temporal calculations (like duration), comparisons (such as length), or pinpoint events at specific times within the video.

**Constrained OCR**

- These questions aim to test the model's text recognition capabilities under specific constraints (like artistic lettering or particular fonts), using a provided image that showcases a specific text style or example. The task involves searching for and extracting text content from the video that matches the specified style.

**Attribute Change**

- These questions aim to test the model's ability, using the provided image (which typically designates the object/person to track), to continuously follow that specific target across different segments of the video and analyze/describe how its attributes (e.g., color, state, location) change over time.

**Temporal Reasoning**

- These questions aim to test the model's ability, using the provided image (which might reference event types, participants, or scenes), to understand the sequence of recurring events within the video and perform temporal reasoning to locate the specific time point or time interval corresponding to the Nth occurrence of a particular event in that sequence.

### A.1.2 DATA ANNOTATION STEPS

**🔧 Operations**

**Watch Video and Determine Question Type:**

- After watching the video content to be annotated in its entirety, select the most suitable and valuable question type from the 14 pre-defined types based on the video content, and then brainstorm the question direction accordingly to prepare for subsequent question stem and answer design.

**Design Question Stem and Answer:**

- **Question Stem Design Requirements:**
    - **Close Relevance:** Ensure the question stem is closely related to the content of the video and paired images.
    - **Information Confidentiality:** The question stem should avoid revealing any direct information about the video and images, retaining only necessary hints.
    - **Assessment Significance:** Question stem design should have assessment significance, avoiding simple questions of purely objective facts.
    - **Concise and Clear Language:** Use concise and clear language to describe the question, avoiding ambiguity or redundancy.

    **Answer Design Requirements:**
    - **Unique Clarity:** For the question stem, there must be a unique and clearly correct answer.
    - **Information Confidentiality:** The content of the correct answer itself must not directly reveal any information about the paired images.
    - **Video Granularity:** Specify the smallest video unit required to answer the question, such as:

**Perception Tasks**

**Reasoning Tasks**

**Natural Language Inference**

Query Image

Which scene in the <Query Image> matches the style of a scene in the video?
A: Second from the left    D: Second from the right
B: First from the right     E: first from the left
C: first from the left       . . .

**Temporal Reasoning**

Query Image

FOOD

When did the video first fully display the content shown in the <Query Image> ?
A: 01:26-01:28    D: 02:08-02:10
B: 01:24-01:26     E: 01:20-01:22
C: 01:22-01:24     . . .

**Reverse Existence**

Query Image

Which person in the <Query Image> does not appear in the video?
A: First from the left...    D: First from the...
B: Second from the right...   ...
C: Second from the left...

**Counting**

Query Image

THE HATE U GIVE
ANGIE THOMAS

The video blogger recommended a couple of books, in order, which book is the content in the <Query Image> adapted from?
A: 1    D: 4
B: 2    E: 5
C: 3    . . .

**Constrained OCR**

Query Image

What is the artistic text above the image shown in the <Query Image> ?
A: YORK         D: TFW
B: GREATNESS    E: Slay The DRAGON
C: PRFECT       . . .

Figure 5: Five remaining IV-Bench task categories: Natural Language Inference, Constrained OCR, Reverse Existance, Counting, and Temporal Reasoning. Each example requires using text, image, and video together.

       ∗ **Frame:** The answer is located at a specific frame in the video.
       ∗ **Clip:** The answer is located in a continuous clip of the video.
       ∗ **Full Video:** The answer requires understanding the full video content.
   – **Video Range:** Clearly indicate the specific segment range in the video where the answer is located to quickly verify the accuracy of the answer.

COLLECT PAIRED IMAGES:

- **Diversity of Image Sources:** Widely collect images that meet the question requirements from the internet or video resources, avoiding single and over-reused image sources to ensure image diversity.

- **Non-Video Screenshots:** Directly capturing frames from the current test video as paired images is prohibited.

IMAGE QUALITY ASSURANCE:

- **Texture Clarity:** The texture of the image must be clearly distinguishable, avoiding blurriness to ensure the effectiveness of visual information.

- **Subject Consistency:** The characters or objects in the image must be completely consistent with or visually highly similar in texture to the characters or objects appearing in the video.

- **Subject Prominence:** The target object in the image should occupy the main position of the image, highlighting the subject and reducing background interference for easy observation and identification.

- **Visual Information Validity:** Images must meet pre-defined "visual information validity requirements."

- **Annotation Validity Requirement Number:** Based on the specific basis for selecting images, annotate the corresponding number (requirement_number) of the "visual information validity requirements" that the image meets. (Please provide a pre-defined list of "visual information validity requirement" numbers for accurate annotation).

DESIGN DISTRACTOR OPTIONS (9 OPTIONS):

- **Number of Distractor Options:** Each question needs to design 9 misleading incorrect options, plus 1 correct answer, for a total of 10 options.
- **Diversified Confusion Strategies:** Avoid patterned option design. Incorrect answers should be set from different angles and dimensions to increase the discrimination and difficulty of the questions. Common confusion strategies include:
  - **Conceptual Confusion:** Use options with concepts similar to the correct answer but with subtle differences in meaning to test the precise understanding of concepts.
  - **Partial Correctness:** The content described in the option partially matches the video content, but does not fully answer the question raised in the question stem.
  - **Irrelevant Information:** The option content has low direct relevance to the video content, but may have some connection with the question in daily cognition or common sense, setting up interference.
  - **Incorrect Reasoning:** Conclusions obtained from logical reasoning that seems reasonable but is actually incorrect based on the video content are used as distractor options.
  - **Image Misdirection:** Use objective facts or visual information presented in paired images to design options that match the images but not the video, forming interference.
  - **Conclusion Divergence:** For the same question stem, change different paired images to make the conclusion of the options change with the images, and use this as a distractor item.
  - **Real-World Significance Trap:** Design options that have certain significance or rationality in real life based on the images, but do not conform to the video content, inducing users to answer based on common sense rather than the video.
  - **Avoid Fabrication:** Avoid designing options with overly obvious traces of fabrication, ensuring that distractors have a certain degree of misleadingness and avoiding easy exclusion by users.

OPTION FORMAT REQUIREMENTS:

- **Length Consistency:** Ensure that the lengths of the 10 options (including the correct answer and 9 distractors) are approximately equal to avoid answer information leakage due to option length differences.
- **Concise Language:** The language expression of distractor options should be concise and clear, avoiding unnecessary complex sentence structures and rare vocabulary. At the same time, the descriptive language of distractor options should avoid revealing any information about the paired images.

### A.1.3 METHODS FOR DESIGNING DISTRACTORS

🔧 Distractors Design Methods

- **Visual Replacement:** Replace a visual element in the video (such as the color, shape, or texture of an item) with visual information that is similar but inaccurate to the actual content.
- **Quantitative Replacement:** Replace a numerical detail in the video (such as quantity, time, distance, etc.) with an incorrect numerical value.
- **Spatial Replacement:** Incorrectly describe the location where an event occurs, for example, misdescribing "in the kitchen" as "in the living room" or another space.
- **Temporal Replacement:** Incorrectly describe the time point when an event occurs, for example, misdescribing "morning" as "evening."
- **Addition of Information:** Deliberately add non-existent events or information to the video content, such as fabricating a plot or detail.
- **Missing Information:** Delete important information or events that exist in the video, for example, deliberately omitting key information, leading to incomplete information.
- **Detail Replacement:** Incorrectly replace key information involving characters, events, or details in the video, for example, replacing the profession or age attributes of a character, or incorrectly describing the details of an event.

- **Sequential Replacement:** Arrange a series of actions or events that actually occurred in the video in the wrong order, disrupting their chronological relationship.

- **Reality Trap:** Based on people's common sense or logic in real life, design options that have a certain meaning or rationality in reality, but these options do not match the actual content of the video.

- **Conclusion Divergence:** For the same video content and question stem, by changing different paired images, the conclusion of the options changes with the paired images.

---

### 🔧 Video Content Guidelines and Question Screening Process

**Video Content Guidelines:**

- **Include Similar Distractors:**
  - The video must include at least two figures or objects of the same type as the image target to create visual confusion.
- **Meet any of the following conditions to emphasize the importance of texture:**
  - **Condition 1: Texture-Dominant Definition**
    * The image target can only be identified in the video through texture characteristics.
    * *For example: Only present close-up texture details of a human face, weakening features like contours and clothing.*
  - **Condition 2: Key Feature Differentiation**
    * Distractors of the same type in the video differ from the image target in at least one key visual feature.
    * *For example: Shoes of the same style but different colors, the same person wearing different styles of clothing.*
  - **Condition 3: Multiple Feature Description Requirement**
    * The image target requires at least four visual features to be fully described.
    * Emphasize the complexity of the target's visual information, requiring multi-dimensional features for accurate understanding.

**Question Screening Process:**

- **Generate Detailed Description:**
  - After annotation, use MLLM to generate a Detailed Description based on the image and question, controlling the granularity of the text description.
- **Assess Answerability:**
  - Use MLLM or human evaluation, combined with the Detailed Description + video, to attempt to answer the question.
- **Question Screening:**
  - Eliminate questions that can be answered correctly based solely on the Detailed Description + video.
  - Retain questions to ensure that image texture information is crucial for a correct answer.

---

## A.2 QUALITY CONTROL PROCESS DETAILS

The quality inspection of IV-Benchmark comprises two rounds:

- **Round 1:** focuses on standardizing problem structures and content validity.

- **Round 2:** addresses advanced quality requirements to ensure task rigor.

### A.2.1 ROUND 1 QUALITY CONTROL

> **⊙ Purpose**
>
> - Ensure basic structural integrity and content standardization, including unambiguous question formulation, verifiable answers, reasonable distractors, and data quality.

> **ℹ Quality Assessment Dimensions**
>
> - **Clarity Validation:** Verify grammatical correctness and unambiguous expression of questions/options.
> - **Answer Validity Validation:** Confirm answers are deducible from video content (eliminate labeling errors).
> - **Task Categorization Calibration:** Validate proper classification of question types.
> - **Contextual Validation :** Contextual Validation: Ensuring answers and distractors are plausible and contextually relevant.
> - **Image Quality Assurance:** Check query image resolution and visibility of critical information.
> - **Option Completeness Validation:** Verify coverage of plausible alternatives (e.g., critical missing options in multi-choice questions).

### A.2.2 ROUND 2 QUALITY CONTROL

> **⊙ Purpose**
>
> - Ensure task validity by verifying the necessity of multimodal components and the contextual plausibility of distractors, thereby mitigating evaluation bias caused by design flaws.

> **ℹ Methods**
>
> - **Multimodal Necessity Check:** Retain only questions requiring combined analysis of text/image/video.
> - **Information Leakage Detection:** Identify text queries that inadvertently reveal visual content through textual cues (e.g., explicit object descriptions, positional references) and eliminate leakage by rewriting queries to preserve task intent.
> - **Commonsense Dependency Screening:** Eliminate questions answerable through general knowledge alone (e.g., "the sun rises in the east").
> - **Distractor Optimization:** Redesign meaningless distractors based on video content. Preferred: Distractors should match answer categories or create confusion (e.g., use actual OCR text from videos for text-related questions).
> - **Object Uniqueness Verification:** Ensure non-unique targets in questions (e.g., "What color is the person's clothing shown in both image and video?" requires multiple persons in video).

Quality control was primarily the responsibility of our author team and conducted in multiple stages. These stages encompassed strict manual checks against all defined quality standards and requirements. The entire process was predominantly manual; we relied on expert judgment for all final quality decisions, ensuring data validity and accuracy. Although we explored auxiliary tools, the ultimate quality assessment and decision-making entirely depended on our human expertise. This direct, manual oversight was key to guaranteeing the quality and complexity of the benchmark.

### A.2.3 STATISTICAL ANALYSIS OF QUALITY CONTROL

To quantify the rigor of our manual review process and ensure the reliability of IV-Bench, we implement a "1-Generation, 3-Verification" pipeline. This involves tracking retention rates at each stage and calculating inter-annotator agreement among experts.

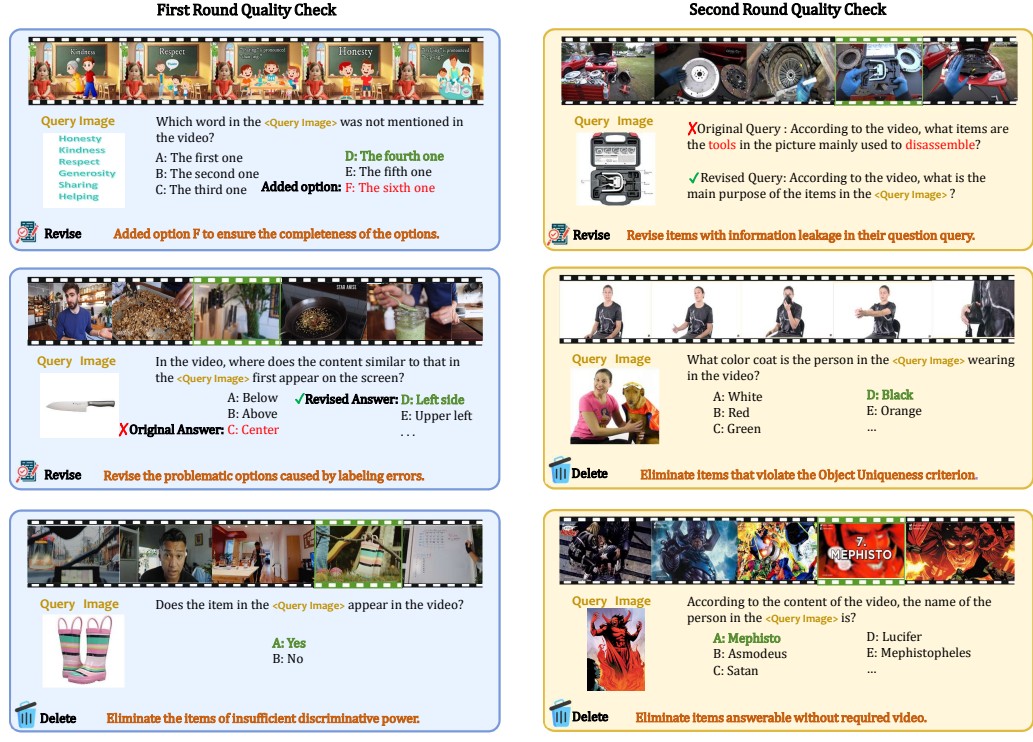

Figure 6: Representative data examples from Two Round Quality Check. Each round includes modifications and deletions to queries and images.

**Pipeline Statistics and Retention Rates.** Our multi-stage filtering process acts as a funnel to ensure only high-quality samples remain. We begin with an initial pool of 1,500 videos and approximately 4,500 questions.

- **Round 1 (Standardization):** We filter out 285 videos and 1,072 questions due to structural flaws or ambiguity, resulting in a retention rate of 81.0% for videos.
- **Round 2 (Leakage & Necessity):** We further remove 249 videos and 868 questions that fail strict multi-modal necessity checks or contain unfixable leakage.

The final benchmark comprises **966 videos** and **2,560 questions**. As detailed in Table 3, the overall retention rates are 64.4% for videos and 56.9% for questions, reflecting our high standard for data validity.

Table 3: **Statistics of the Quality Control Pipeline.** The rigorous filtering process ensures that only high-quality, non-leaking, and strictly image-grounded samples remain.

| Stage | Videos | | | Questions | | |
|---|---|---|---|---|---|---|
| | Count | Filtered | Retention | Count | Filtered | Retention |
| Initial Pool | 1,500 | - | - | 4,500 | - | - |
| Round 1 (Standardization) | 1,215 | 285 | 81.0% | 3,428 | 1,072 | 76.2% |
| Round 2 (Leakage & Necessity) | **966** | 249 | 79.5% | **2,560** | 868 | 74.7% |
| **Overall Retention** | - | 534 | **64.4%** | - | 1,940 | **56.9%** |

**Inter-Annotator Agreement.** To ensure consistency in our expert judgment (particularly for subjective criteria like "distractor effectiveness" and "image necessity"), we calculate the agreement among the three independent inspectors for each verification decision. We compute **Fleiss' Kappa**

across the validation process and achieve a score of:

$$\kappa = 0.86 \tag{1}$$

This value indicates **strong inter-annotator agreement**, validating that our quality control standards are applied objectively and consistently across the dataset.

### A.3 INFERNCE PROMPT

Multiple-choice questions are constructed by pairing each instance with one correct answer and several distractors. During inference, the answer choices are randomly shuffled to ensure that the correct answer appears in different positions. The default placement order is video, image, and text. For some models, we also experiment with placing the image after the video. Therefore, we have two different text prompts: one with the order of video, image, and text, and another with the image placed after the video. In the prompt, we explicitly specify the order of the video and image, the video length, and instruct the model to answer the question based on both the video and the image, providing only the options.

> **ⓘ Prompt**
>
> - **Video-first prompt:** <VIDEO> <IMAGE> We provide you with an image placed at the very beginning, followed by a video that has been divided into `evenly spaced frames` across its `seconds duration`. Please answer the question based on the content from both the image and the extracted video frames.
> - **Image-first prompt:** <IMAGE> <VIDEO> We provide you with a video that has been divided into `frame_num` evenly spaced frames across its `duration` seconds duration, followed by an image. Please answer the question based on the content from both the video frames and the image.

### A.4 RESOURCE

All experiments on open-source MLLMs are conducted on cloud servers with NVIDIA A800 80G GPUs and Intel Xeon Platinum 8336C @ 2.30 GHz CPUs. For open source MLLMs, we obtain their results where their API services. To avoid unknown impacts on results, we do not conduct batched inference and set batch size=1 for all MLLMs.

### A.5 JUSTIFICATION FOR IMAGE-GROUNDED VIDEO PERCEPTION AND REASONING

The introduction of our benchmark for Image-Grounded Video Perception and Reasoning is not an isolated academic pursuit but a direct response to a convergence of massive, real-world user behaviors and critical backend technical needs. This appendix provides a comprehensive justification for the task's necessity, demonstrating that it is grounded in both large-scale, commercially vital user habits and a range of established and emerging technological applications.

#### A.5.1 THE USER-DRIVEN MANDATE: IMAGE QUERIES AS A DOMINANT BEHAVIOR

The necessity of understanding visual queries is unequivocally demonstrated by their massive scale in production systems and clear user preference. This is not an anecdotal use-case but a foundational aspect of modern digital interaction.

- **Massive Scale and Commercial Vitality:** User-facing evidence confirms that visual search is a large-scale and commercially vital behavior. As early as 2018, Pinterest's Lens platform was handling **600 million image searches per month** Zhai et al. (2017), while Alibaba's image search application grew to over **17 million daily active users** Zhang et al. (2018). These figures establish an undeniable foundation for the necessity of models that can ground their understanding in a user-provided image.
- **Strong User Preference:** This massive adoption is driven by strong user sentiment. Research reports that **62% of Millennial and Gen Z consumers wish for visual search**

**capabilities**, and **85% of young respondents place more importance on visual information than text** when shopping. This preference is rooted in visual search's unique ability to handle queries that are difficult to verbalize Dagan et al. (2023). As a "key driver for user engagement" that provides "a significant lift in key business metrics" Zhai et al. (2017), the evidence is clear: image-grounded interaction is a core user expectation.

### A.5.2 THE TECHNICAL IMPERATIVE: IMAGE-GROUNDED UNDERSTANDING AS A CORE CAPABILITY

The large-scale user demand for visual queries fuels a wide range of backend technical applications where image-grounded video understanding is not just beneficial but essential. The advent of MLLMs has significantly enhanced performance in these areas, making them more prevalent. Key applications include:

- **Video Editing and Creation:** To generate a relevant video from given multimodal inputs, such as in all-in-one video creation and editing platforms, a robust joint understanding of images and videos is required Jiang et al. (2025). For instance, in script-based movie generation, visual information about main subjects or backgrounds must first be acquired from images and then used for co-reference throughout the script, demanding a comprehensive integration of image and video contexts Wu et al. (2025b).

- **Image-to-Video Generation:** This task requires an MLLM to comprehend the relationship between static frames and the complete video dynamic, which is critical for creating coherent image-text-video triplets for training Yang et al. (2025). This core capability is essential across various sub-tasks, including image animation, which animates a subject while preserving identity Tan et al. (2024); Xu et al. (2024), and virtual try-on for E-commerce, which demands a precise understanding of image details to prevent distortion Fang et al. (2024).

- **Image-based Video Retrieval:** Video retrieval is a cornerstone for training specialized MLLMs, video generation models, and recommendation systems. When the desired content is difficult to describe with text, direct image-based retrieval offers a superior alternative. While historically based on embedding methods Bolya et al. (2025), the rapid advancement of RAG technology has led to a new wave of multimodal query-based video retrieval systems that require more sophisticated image-grounded reasoning Ventura et al. (2024); Thawakar et al. (2024).

- **Image-Aided Video Understanding:** A mainstream approach to long-video understanding involves first identifying keyframes or crucial visual elements from the video. These extracted visual cues (images) are then fed into an MLLM alongside the original video for a more focused and comprehensive analysis Han et al. (2025); Luo et al. (2024); Ma et al. (2025). This concept is evolving into more complex reasoning tasks, with a growing body of work incorporating the idea of "thinking with images" directly into the video understanding process Zhang et al. (2025c).

In conclusion, the combination of overwhelming user demand for visual interaction and the foundational role of image-grounded understanding across numerous advanced AI applications provides a powerful and timely justification for our work. The ability to perceive and reason about video content based on an external image is not a niche capability but a central challenge for the next generation of multimodal systems.

### A.5.3 QUANTITATIVE ANALYSIS ON THE SEMANTIC GAP AND IMAGE NECESSITY

While the sections above establish the broad utility of image-grounded video understanding, we further conduct a comprehensive quantitative analysis to rigorously justify the structural design of IV-Bench. Specifically, we demonstrate why *external* images with diverse semantic gaps—ranging from near-frame extraction to high-level semantic abstraction—are essential for a robust evaluation.

**Experimental Setup.** We quantify the "semantic gap" by calculating the cosine similarity between the external image query and 8 uniformly sampled video frames using CLIP-ViT-L/14. Based on the similarity scores across 2,560 samples, we divide the dataset into three groups:

- **High-Similarity (Top 25%,** $> 0.567$**):** Scenarios akin to near-frame extraction or direct visual matching.

- **Mid-Similarity (Mid 50%,** $0.427$–$0.567$**):** Scenarios with a moderate semantic gap.

- **Low-Similarity (Bottom 25%,** $< 0.427$**):** Scenarios with a high semantic gap, requiring strong reasoning.

**Insight 1: Diverse Gaps are Essential for Comprehensive Evaluation.** We evaluate 10 state-of-the-art MLLMs across these groups (Table 4). The results reveal divergent model behaviors that a single-type dataset fails to capture. Some models (e.g., InternVL3.5-8B-MPO) show a clear preference for **High-Sim** scenarios ($+3.29\%$), indicating strong visual pattern matching but weaker semantic generalization. Conversely, models like Qwen3-VL-32B excel in **Low-Sim** scenarios ($-6.58\%$ diff), demonstrating superior capability in handling abstract semantic connections. This underscores that a holistic assessment must include diverse semantic gaps to evaluate both fine-grained visual correspondence and semantic reasoning capabilities.

Table 4: **Model performance across semantic gaps.** A positive difference denotes a preference for High-Sim (Near-Frame) inputs, while a negative difference denotes better performance on Low-Sim (Reasoning) inputs.

| Model | High-Sim | Mid-Sim | Low-Sim | Diff (H-L) |
|---|---|---|---|---|
| Aria-32B | 19.59 | 15.22 | 19.91 | -0.31 |
| Qwen2.5-VL-7B | 20.85 | 16.71 | 19.28 | +1.57 |
| Qwen3-VL-8B | 19.28 | 16.71 | 20.22 | -0.94 |
| Qwen3-VL-32B | 20.53 | 23.29 | 27.12 | **-6.58** |
| InternVL2.5-26B | 22.10 | 19.29 | 21.94 | +0.16 |
| InternVL3-8B | 21.94 | 16.63 | 20.69 | +1.25 |
| InternVL3-14B | 25.39 | 24.47 | 27.12 | -1.72 |
| InternVL3-38B | 26.49 | 24.47 | 27.43 | -0.94 |
| InternVL3.5-8B | 21.79 | 17.80 | 19.75 | +2.04 |
| InternVL3.5-8B-MPO | 22.26 | 18.35 | 18.97 | **+3.29** |

**Insight 2: Task-Specific Dependencies and Discriminative Power.** The necessity of diverse images is further corroborated by task-level analysis (Table 5). Tasks relying on direct visual verification (e.g., *Detailed Events*, *Existence*) benefit significantly from high visual similarity ($+20.64\%$ gap). However, reasoning-intensive tasks (e.g., *Attribute Change*, *Natural Language Inference*) perform substantially better with larger semantic gaps. Notably, *Attribute Change* achieves $10.47\%$ higher accuracy in low-similarity settings. This suggests that high-similarity images may introduce bias by fixing attention on static states, whereas external images with a larger semantic gap encourage the model to focus on attribute transformations.

Furthermore, low-similarity samples demonstrate a **39.7% higher standard deviation** in model performance compared to high-similarity samples, indicating that scenarios with larger semantic gaps provide stronger differentiation power for evaluating advanced model capabilities.

## A.6 DIVERSITY ANALYSIS OF EXTERNAL IMAGES

To ensure that IV-Bench robustly evaluates MLLMs across a wide spectrum of real-world scenarios, the external grounding images were curated to exhibit substantial diversity in both content and complexity. This section provides a quantitative analysis of this diversity.

### A.6.1 METHODOLOGY FOR CLASSIFICATION

To systematically categorize the visual content, we recruited three annotators with significant experience in evaluating Multimodal Large Language Models (MLLMs) to classify all 2,560 external images into seven distinct categories based on their primary subject matter. This classification provides a clear overview of the visual contexts represented in the benchmark, confirming that models are tested against a varied set of grounding inputs.

Table 5: **Performance by Task Type across Semantic Gaps.** Tasks are sorted by their dependency on high visual similarity (Diff H-L).

| Task Type | High-Sim | Mid-Sim | Low-Sim | Diff (H-L) |
|---|---|---|---|---|
| Detailed Events | 35.37% | 21.44% | 14.73% | +20.64% |
| Summary | 26.38% | 18.28% | 16.39% | +9.99% |
| Instruction Understanding | 17.96% | 21.86% | 11.62% | +6.34% |
| Existence | 20.00% | 15.70% | 16.18% | +3.82% |
| Reverse Existence | 16.51% | 14.70% | 13.39% | +3.12% |
| Keyframe Extraction | 16.81% | 14.05% | 13.93% | +2.88% |
| Counting | 20.45% | 18.11% | 19.66% | +0.80% |
| Space-Time Computing | 18.53% | 12.95% | 18.18% | +0.35% |
| Temporal Reasoning | 12.03% | 11.55% | 12.50% | -0.47% |
| Spatial Relationship | 32.66% | 25.52% | 34.85% | -2.19% |
| Constrained OCR | 31.49% | 29.55% | 36.64% | -5.16% |
| Natural Language Inference | 28.57% | 28.89% | 36.36% | -7.79% |
| Attribute Change | 18.02% | 22.48% | 28.48% | -10.47% |

### A.6.2 DISTRIBUTION OF IMAGE CATEGORIES

The distribution of the external images across the seven content-based categories is detailed in Table 6. The results show a balanced yet comprehensive collection, with a strong representation of both object-centric and person-centric scenes.

Table 6: Distribution of external images across seven content-based categories. The variety ensures that models are evaluated on a wide range of visual grounding scenarios.

| Category | Single Person | Multi Person | Single Object | Multi Object | Landscape | Text-Dominant | Others |
|---|---|---|---|---|---|---|---|
| **Count** | 553 | 293 | 792 | 296 | 187 | 303 | 136 |

As the data illustrates, these seven categories span a broad range of visual contexts. They range from simple compositions like single-person portraits and isolated object shots to more complex scenarios involving multi-person and multi-object scenes, as well as ambient landscapes and information-rich, text-dominant images. This variety in both structural and semantic content ensures that IV-Bench's external images rigorously challenge a model's image-grounded video perception and reasoning capabilities. By preventing models from overfitting to a narrow type of visual query, this diversity promotes the development of more generalized and robust systems.

### A.7 ABLATION STUDY ON IMAGE VS. TEXTUAL CAPTION GROUNDING

To validate the necessity of using external images for grounding in IV-Bench, we conduct an ablation study to investigate whether the proposed tasks can be simplified into a text-only video question-answering problem. Specifically, we evaluate a *naive* two-step baseline where the external image is first converted into a machine-generated textual description (a caption), which then serves as the textual context alongside the video.

### A.7.1 EXPERIMENTAL SETUP

We evaluate this approach on two powerful open-source models, Qwen2.5-VL-32B and Qwen2.5-VL-72B, under three distinct settings:

1) **Video-Only:** The model receives only the video frames and the text query, with no image-based context. This serves as our lower-bound baseline.

2) **Video + Caption:** The external image is replaced by an auto-generated caption. To ensure a fair comparison, the caption for each image is generated by the model under evaluation itself.

3) **Video + Image:** The model is provided with the full input triplet: video, the raw external image, and the text query. This represents the standard setting for IV-Bench.

Consistent with the main paper's evaluation protocol, all experiments are conducted using 32 video frames at a 480p resolution.

### A.7.2 RESULTS AND ANALYSIS

The performance comparison across the three settings is summarized in Table 7.

Table 7: Performance comparison of different input modalities on IV-Bench. The Video + Caption setting demonstrates that while textual descriptions of the image provide some benefit over a Video-Only baseline, they are significantly outperformed by the direct use of the raw image (Video + Image).

| Model | Video-Only | Video + Caption | Video + Image |
|---|---|---|---|
| Qwen2.5-VL-32B | 18.0% | 20.9% (+2.9%) | **23.0%** |
| Qwen2.5-VL-72B | 18.6% | 24.8% (+6.2%) | **28.9%** |

The results clearly indicate that while adding a textual caption of the image provides a notable performance uplift over the Video-Only baseline (+2.9% for the 32B model and +6.2% for the 72B model), it is insufficient to match the performance of true image grounding. A significant performance gap persists between the Video + Caption and Video + Image conditions (2.1% for 32B and 4.1% for 72B).

This persistent gap underscores a core motivation for IV-Bench: textual summaries inevitably omit critical visual information that is essential for correct answers. Nuances such as *fine-grained object attributes*, *specific human facial features*, and *complex spatial details* are often lost during the captioning process. Therefore, although captioning can be considered a strong *naive* baseline, only direct image input can fully capture the depth of visual perception and reasoning that our benchmark is designed to evaluate. This experiment confirms that the challenges posed by IV-Bench cannot be circumvented by relying on text-only shortcuts and require genuine multimodal reasoning grounded in direct visual evidence.

### A.8 HUMAN TEST

To explore the gap between Multimodal Large Language Models (MLLMs) and human performance on IV-Bench, we recruit ten experienced annotators who, over four days, answer all IV-Bench questions. To facilitate a comprehensive understanding and accurate responses, we provide the annotators with the complete video, corresponding images, and the full text associated with each question. The evaluation is conducted under a strict protocol where we instruct annotators to base their answers solely on the provided materials, without the use of any external tools such as search engines or other aids. The average of the human performance is then calculated and included in our paper for comparison with the model's performance.

### A.9 ROOT CAUSE ANALYSIS: INFORMATION AVAILABILITY VS. TEMPORAL CAPABILITY

A significant observation from our main evaluation is the limited performance on frame-sensitive tasks, such as *Temporal Reasoning*, *Counting*, and *Space-Time Computing*. A natural hypothesis is that this limitation stems from sparse frame sampling (typically 8–32 frames), which may miss critical temporal details required for fine-grained understanding.

To investigate whether the bottleneck lies in insufficient visual information (i.e., the sampling rate) or inherent model capabilities, we conducted a controlled experiment using **Qwen3-VL-8B**. We compared performance under a standard setting (32 frames) versus a high-density setting (128 frames) to isolate the impact of information availability.

**Experimental Setup: Quantifying Information Gain.** First, we verified that increasing the frame count indeed provides more valid visual information. We define two metrics:

- **Temporal Coverage:** The percentage of samples where at least one sampled frame falls within the ground truth (GT) relevant time segment.

- **Frame Density:** The average number of frames sampled per second.

As shown in Table 8, increasing frames from 32 to 128 results in a substantial increase in information availability, with Temporal Coverage improving by nearly 10 percentage points and Frame Density increasing by $3.2\times$.

Table 8: Comparison of information availability between 32 and 128 frames.

| Metric | 32 Frames | 128 Frames | Improvement |
|---|---|---|---|
| Temporal Coverage | 76.09% | 85.94% | +9.85 pp (+13.0%) |
| Frame Density | 0.13 fps | 0.41 fps | +3.2× (+216%) |

**Performance Analysis: The Divergence.** We focused our analysis on four frame-sensitive tasks that exhibited similarly low baseline performance ($< 20\%$). The results, presented in Table 9, reveal a critical divergence in how models respond to increased information.

Table 9: Performance comparison on frame-sensitive tasks using Qwen3-VL-8B.

| Task Category | 32 Frames | 128 Frames | Change |
|---|---|---|---|
| Counting | 14.98% | 16.48% | +1.50% |
| Space-Time Computing | 14.02% | 16.36% | +2.34% |
| Keyframe Extraction | 18.47% | 17.77% | -0.70% |
| Temporal Reasoning | 13.33% | 11.11% | -2.22% |

While tasks relying on information aggregation (e.g., *Counting*, *Space-Time Computing*) benefit modestly from the increased frame count, performance on *Temporal Reasoning* notably declines.

**Discussion: The 'Static Bias' Bottleneck.** This paradox—where more relevant visual information leads to worse reasoning performance—strongly suggests that the primary bottleneck is not the lack of visual cues, but rather an inherent deficiency in the model's temporal modeling mechanism.

1. **Information Overload vs. Utilization:** If the limitation were simply missing frames, *Temporal Reasoning* should have improved alongside *Counting*. The decline implies that the additional frames introduced "information overload" rather than useful signals for sequential logic.

2. **Static Bias:** Our findings align with the concept of "Static Bias" identified in recent studies (e.g., MVBench Li et al. (2024c)). Current MLLMs predominantly rely on static image encoders (e.g., CLIP) trained to align single images with text. This design encourages the model to process video as a "bag of frames," effectively averaging static content rather than modeling temporal flow or causality.

3. **Temporal-Spatial Performance Gap:** Consequently, we observe a significant gap where increasing frame density fails to bridge the reasoning deficit. Without explicit mechanisms to model inter-frame dynamics, simply feeding more frames exacerbates the noise for reasoning tasks, highlighting the urgent need for architectures that move beyond static visual encoding to genuine causal modeling.

## A.10 SYMMETRIC IMPACT OF MODEL SCALING: PERCEPTION VS. REASONING

To investigate whether simply increasing model capacity solves the challenges in image-grounded video understanding, we conduct a detailed analysis of scaling effects. We categorize the 13 tasks into **Perception Tasks** (e.g., *Existence, Detailed Events*) and **Reasoning Tasks** (e.g., *Counting, Temporal Reasoning*) and evaluate performance gains across five major model families.

**Experimental Results.** Table 10 summarizes the performance of models ranging from 7B to 78B parameters. We observe a consistent and significant divergence in how scaling benefits different task types.

Table 10: **Scaling Effects on Perception vs. Reasoning.** Larger models achieve massive gains in perception but show diminishing returns in reasoning. (Gain denotes the absolute improvement from the smallest to the largest model in the family).

| Model Family & Size | Overall | Perception | Reasoning |
|---|---|---|---|
| *Qwen2.5-VL Series* | | | |
| Qwen2.5-VL-7B | 18.5% | 18.9% | 17.9% |
| Qwen2.5-VL-32B | 23.0% | 24.9% | 20.2% |
| Qwen2.5-VL-72B | 28.9% | 33.7% | 21.9% |
| **Gain (7B → 72B)** | **+10.4%** | **+14.8%** | **+4.0%** |
| *Qwen3-VL Series* | | | |
| Qwen3-VL-8B | 18.2% | 20.1% | 15.4% |
| Qwen3-VL-32B | 23.5% | 27.7% | 17.4% |
| **Gain (8B → 32B)** | **+5.3%** | **+7.6%** | **+2.0%** |
| *InternVL2.5 Series* | | | |
| InternVL2.5-8B | 17.4% | 17.8% | 16.5% |
| InternVL2.5-26B | 20.6% | 22.4% | 18.1% |
| InternVL2.5-38B | 26.6% | 30.4% | 21.1% |
| InternVL2.5-78B | 28.6% | 33.4% | 21.9% |
| **Gain (8B → 78B)** | **+11.2%** | **+15.6%** | **+5.4%** |
| *InternVL3 Series* | | | |
| InternVL3-8B | 18.9% | 20.0% | 17.3% |
| InternVL3-14B | 25.3% | 27.2% | 22.5% |
| InternVL3-38B | 25.6% | 27.2% | 23.4% |
| **Gain (8B → 38B)** | **+6.7%** | **+7.2%** | **+6.1%** |
| *InternVL3.5 Series* | | | |
| InternVL3.5-8B | 19.3% | 18.9% | 19.8% |
| InternVL3.5-38B | 26.0% | 27.3% | 24.0% |
| **Gain (8B → 38B)** | **+6.7%** | **+8.4%** | **+4.2%** |

**Analysis: Perception Dominates, Reasoning Lags.** Our analysis reveals two critical trends:

1. **Perception Dominates Scaling Gains:** Across almost all model families, perception tasks witness massive improvements as model size increases. For instance, the Qwen2.5-VL series sees a **+14.8%** jump, and InternVL2.5 sees **+15.6%**. This confirms that increasing parameters significantly enhances the model's capacity to "see," identify, and match static visual content.

2. **Reasoning Encounter Diminishing Returns:** In sharp contrast, reasoning tasks show much more modest gains (e.g., only **+4.0%** for Qwen2.5-VL and **+2.0%** for Qwen3-VL). Even the largest 70B+ models struggle to break past the 22–23% accuracy mark on reasoning tasks, despite their strong perception capabilities ($> 33\%$).

**Implication.** This finding highlights a critical limitation of current scaling laws in the context of IV-Bench. While scaling effectively improves visual recognition, it encounters severe diminishing returns for complex reasoning. This suggests that future advancements must look beyond simple parameter scaling and focus on architectural innovations (e.g., temporal causal modeling) or data-centric approaches (e.g., reasoning-specific curriculum learning) to bridge this gap.

## A.11 IMPACT OF TEXT ORDERING ON IV-BENCH

Given the complex multimodal context in IV-Bench (Video + External Image + Text), the sequential ordering of inputs plays a crucial role. While our default setting follows the widely adopted "Vision-then-Text" paradigm (i.e., Video → Image → Question), we conduct a controlled experiment to investigate the sensitivity of MLLMs to inverted text ordering (i.e., Question → Video → Image).

**Experimental Setup.** We evaluate two distinct model families, MiniCPM and InternVL2.5, under two input configurations:

1. **Text After Vision (Standard):** The standard training configuration for most recent MLLMs Chen et al. (2024c); Li et al. (2024a); Wang et al. (2024), where the textual query appears at the end of the sequence.

2. **Text Before Vision (Inverted):** The textual query is prepended to the visual tokens, forcing the model to retain textual intent across the long visual sequence.

**Results and Analysis.** As shown in Table 11, inverting the text order leads to substantial performance degradation across all tested models.

Table 11: **Impact of Text Ordering.** Placing the text query before visual content results in significant performance drops, highlighting the sensitivity of current MLLMs to input sequence ordering.

| Model | Text After Vision | Text Before Vision | Performance Drop |
|---|---|---|---|
| MiniCPM-V | 17.2% | 13.4% | -3.8% |
| MiniCPM-o | 17.1% | 14.1% | -3.0% |
| InternVL2.5-38B | 26.6% | 15.9% | -10.7% |
| InternVL2.5-78B | 28.6% | 16.5% | **-12.1%** |

**Key Observations.**

- **Universal Degradation:** Placing text before vision causes accuracy drops ranging from 3.0% to 12.1%. This confirms that the relative positioning of textual and visual modalities is not trivial and significantly impacts multimodal integration.

- **Model Sensitivity:** Different architectures exhibit varying degrees of robustness. The InternVL2.5 series is particularly sensitive, with drops exceeding 10%, whereas MiniCPM shows relatively better stability.

- **Distribution Shift and Attention Dilution:** We attribute this degradation to two primary factors. First, the "Text Before Vision" order deviates from the standard pre-training objective of most MLLMs (Next Token Prediction based on visual prefix), causing a distribution shift. Second, consistent with the "forgetting" phenomenon discussed in Section A.10, placing text before a lengthy sequence of visual tokens (video frames + image) likely causes attention dilution, where the model under-utilizes the early textual instructions.

These findings empirically justify our choice of the "Vision-then-Text" ordering as the optimal setting for image-grounded video understanding.

A.12 MECHANISM VERIFICATION: ATTENTION SCORE ANALYSIS

To quantitatively investigate the mechanism behind the input ordering effects observed in Section 4.3, we analyzed the attention distribution within the **InternVL3-8B** and **14B** models. We compared two distinct configurations: placing the external image *before* the video frames versus placing it *after* the video frames (closer to the textual query).

**Methodology.** We calculated the attention weights by aggregating scores from all attention heads across all transformer layers during the decoding phase. Specifically, for each generated text token, we computed the average attention weight it assigned to the visual tokens (image vs. video) and averaged these values over the entire IV-Bench dataset.

**Results.** The results, summarized in Table 12, reveal a significant shift in attention distribution based on input ordering.

**Analysis and Key Findings.**

- **Drastic Increase in Image Attention:** When the image is placed *after* the video (closer to the final prediction token), the average attention weight on image tokens increases by

Table 12: **Attention Distribution Analysis.** Placing the image *after* the video frames (closer to the query) drastically increases the attention weight assigned to the image, correlating with improved accuracy.

| Model | Configuration | Accuracy | Image Attn. | Video Attn. |
|-------|---------------|----------|-------------|-------------|
| InternVL3-8B | Image Before Video | 16.19% | 0.0051 | 0.0865 |
|  | Image After Video | **20.21%** | **0.0217** | 0.0730 |
| InternVL3-14B | Image Before Video | 21.73% | 0.0041 | 0.0937 |
|  | Image After Video | **24.60%** | **0.0218** | 0.0781 |

approximately **4× to 5×** (e.g., rising from 0.0051 to 0.0217 for InternVL3-8B). This confirms that proximity to the query significantly enhances the model's ability to attend to the external grounding image.

- **Correlation with Performance:** This surge in image attention strongly correlates with the improvement in accuracy (rising from 16.19% to 20.21% for the 8B model). This empirical evidence suggests that the "Image Before" configuration leads to under-utilization of the grounding signal due to the long context of the intervening video.

- **Verification of Positional Bias:** These statistics empirically verify the "positional bias" or "forgetting" effect. When the image appears early in the context window, it suffers from attention dilution caused by the subsequent video sequence. Placing the image later ensures it remains in the model's active focus, facilitating better visual grounding.

- **Consistency with Sparse Attention:** We also observe that the absolute attention scores for visual tokens are generally low (e.g., ~0.02). This aligns with findings in FastV Chen et al. (2024a), which revealed that visual tokens often receive sparse attention in deeper layers. Our results confirm this sparsity but highlight that even small shifts in this limited budget (from 0.005 to 0.02) can drive significant performance gains.

## A.13 COMPARISON WITH VIDEO-MMMU

Figure 7 shows the main differences between IV-Bench and Video-MMMU. IV-Bench is designed to require all three modalities—text, image, and video—for every question. In contrast, Video-MMMU often allows questions to be answered using only text, making video or image redundant. Specifically:

**Image Necessity**:In Video-MMMU, image-text prompts often include full descriptions that make the image redundant. For instance, the calcium-hydroxide question's text describes both solutions in detail, so viewing the picture of the bottles does not change the answer. In IV-Bench, each image contains distractor objects that are not mentioned in the text. Only by examining the image can one distinguish these distractors—e.g. in the "Which contents..." question, you must look at the picture to see which item never appears in the video.

**Video Necessity**: In Video-MMMU (left of Fig.7), the real-interest-rate question includes both text and video, but it can be answered using only the text, making the video unnecessary. In fact, in the very first Video-MMMU example, neither the image nor the video is needed to find the correct answer. By contrast, in IV-Bench (right of Fig.7), the question "When did the items in the picture first appear in the video?" cannot be answered without watching the video, since only the video reveals the exact timestamp.

**Query Modality**: Video-MMMU uses pure-text for two-thirds of its items: in the Fig.7 "Comprehension" and "Perception" examples (the second row), both queries are text-only, accounting for 2/3 of the dataset. IV-Bench, however, pairs text with an image in every query. For example, in the "Which contents in the image do not appear in the video" item, the model must read the text prompt and inspect the image together to identify the correct distractor.

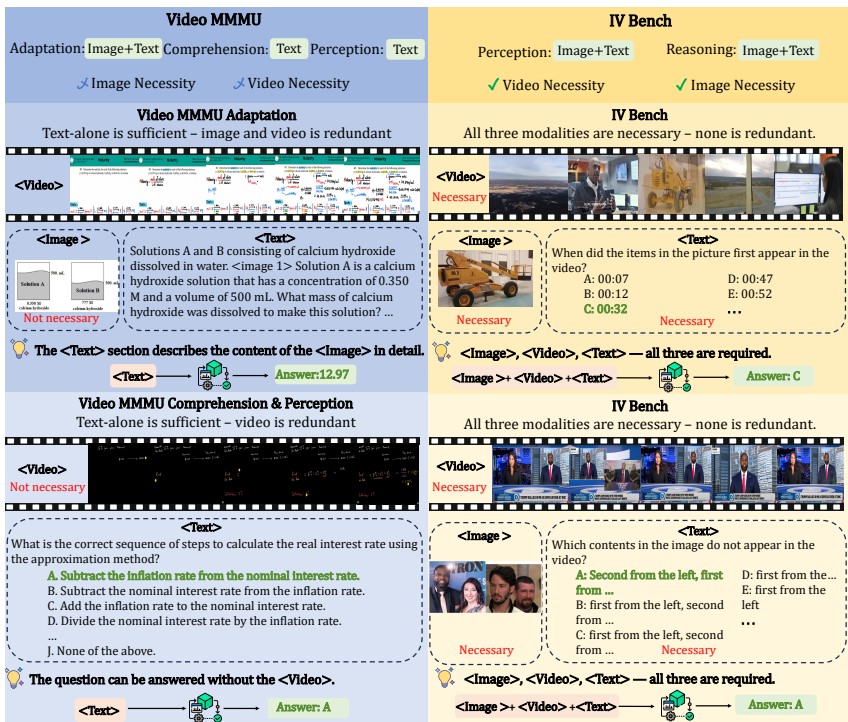

Figure 7: Side-by-side comparison of Video-MMMU (left) and IV-Bench (right). The left side shows that in Video-MMMU many questions can be answered using only text, so the image or video input is often not needed. The right side shows that IV-Bench always requires using the video, image, and text together to answer each question. IV-Bench enforces true multimodal reasoning by adding visual distractors and making each modality necessary, so no single modality (like text alone) can answer the questions.

