# OpenReview forum: "IV-Bench: A Benchmark for Image-Grounded Video Perception and Reasoning in Multimodal LLMs"
_ICLR.cc/2026/Conference — ICLR 2026 Poster_

### Official Review · Reviewer_Zjuc · 2025-10-23

**Soundness:** 3
**Presentation:** 3
**Contribution:** 3
**Rating:** 6
**Confidence:** 3

**Summary:**

This paper presents IV-Bench, the first large-scale benchmark specifically designed to assess image-grounded video reasoning and understanding in multimodal large language models (MLLMs). IV-Bench comprises 966 videos paired with 2,560 manually annotated image–text queries, covering 13 tasks across 5 diverse domains. Through an extensive evaluation of state-of-the-art MLLMs, the study finds that current models achieve no more than 28.9% accuracy, revealing substantial performance gaps.

**Strengths:**

1. The paper tackles an important and previously overlooked problem—image-grounded video understanding—by establishing the first comprehensive benchmark in this area, thereby filling a crucial gap in existing multimodal evaluation benchmarks.

2. The paper highlights the limitations of current MLLMs in handling such tasks, offering clear guidance and future directions for advancing MLLM capabilities.

**Weaknesses:**

The analyses in the paper are not sufficiently deep or insightful. For example, the reported phenomenon of "moderate performance gains with larger models" lacks both theoretical grounding and concrete case studies. Assertions such as "increasing model size primarily enhances memorization and shallow pattern recognition rather than reasoning ability" are not adequately supported by evidence. Similar shortcomings appear throughout other analytical sections. Moreover, the paper fails to propose practical solutions or methodological advances to address the identified challenges.

**Questions:**

I have no questions.

---

> ### Author Response · Authors · 2025-11-24
>
> > ### W1: Insufficient Depth of Analysis and Lack of Solutions
>
> Thank you for this detailed feedback. We appreciate the opportunity to clarify the depth and scope of our analyses. Our conclusions are drawn from extensive experiments across 28 models and 13 task types, representing the most comprehensive evaluation of image-grounded video perception and reasoning to date. As the first benchmark in this domain, IV-Bench enables us to uncover empirical patterns that have not been observable in prior work, and we believe these findings provide substantial insights for the community.
>
> **1. Evidence supporting our analyses.**
>
> Our findings—including model capacity trends, inference-order effects, frame-resolution scaling, and token-budget allocation trade-offs—are supported by consistent patterns across diverse model families (Qwen2.5-VL, InternVL2.5, MiniCPM, LLaVA variants, and others). These trends emerge repeatedly in our data across multiple architectures and task types, as presented in Tables 2-3 and Figures 3-4, providing empirical robustness to our conclusions.
>
> **2. Empirical basis for the perception vs. reasoning observation.**
>
> Regarding the observation about model scaling effects, we would like to clarify the evidence base. Table 2 and our analysis in L368-376 show that in the Qwen2.5-VL series, scaling from 7B to 72B yields substantial improvements on perception-oriented tasks (e.g., Existence: +10.9%, NLI: +8.5%) but more modest gains on reasoning-intensive tasks (e.g., Counting: +2.3%, Temporal Reasoning: +1.8%). This divergent scaling pattern appears consistently across multiple model architectures. While we frame this as an empirical observation specific to image-grounded video understanding rather than a general theory, we believe it offers valuable insights into the limitations of current scaling approaches in this domain.
>
> **3. Actionable insights from our findings.**
>
> We acknowledge that proposing new model architectures is beyond the scope of a benchmark paper. However, our analyses do provide concrete insights that can guide future model development:
>
> - **Input ordering strategy:** Placing the query image after video frames consistently improves performance for larger models (Fig. 3(a)), revealing how positional bias affects multimodal integration.
>
> - **Visual token allocation:** Under fixed token budgets, increasing frame count is more beneficial than increasing resolution, especially for smaller models (Figs. 3(b), 4(a), 4(b)). This offers practical guidance for efficient model deployment.
>
> - **Development priorities:** The divergent scaling on perception vs. reasoning tasks (L368-376) suggests that future innovations might benefit from prioritizing reasoning enhancement mechanisms alongside capacity scaling.
>
> We view our benchmark and findings as establishing a foundation and identifying key challenges, upon which the community can build improved solutions. We hope these insights will inspire future methodological advances in image-grounded video understanding.

---

> > ### Comment · Reviewer_Zjuc · 2025-11-25
> >
> > Thank you for your clarifications. I encourage the authors to present the analyses and key take-aways more explicitly in future revisions. I will increase my scores for soundness and contribution, and will maintain my overall positive rating.

---

### Official Review · Reviewer_2qaq · 2025-10-29

**Soundness:** 4
**Presentation:** 4
**Contribution:** 2
**Rating:** 4
**Confidence:** 5

**Summary:**

This paper proposes a benchmark named IV-Bench that requires the models to perceive and reason the contents from a reference image which is not involved in the input video. The dataset is manually labored and elaborated 7 main findings from evaluating 28 MLLMs.

**Strengths:**

- The curation of the dataset does not rely on automated annotation tools (e.g., GPT-4V) and ensures high quality by manually annotations.
- Assessing model capability to utilize reference images is a novel motivation. Also, excellent presentation with high overall readability and nice figures.
- Impressively extensive evaluation (28 MLLMs) is reported, providing actionable diagnostics. Also, the performance gap between humans and models is huge.

**Weaknesses:**

- The main concern is that the findings are not surprising compared to the quality of the dataset. Comparisons across the model capacity, video fps, and resolution are trivial according to the scaling law [1].
- Although it is not necessary, the analysis lacks experiments on meta-data (e.g., subtitles).
- All tasks are based on multi-choice questions, not requiring open-ended questions even for the “reasoning” category.
- Inter-annotator agreement and quality-control rejection rates on both rounds are missing.
- The paper rigorously describes the model performance by model families (e.g., line 349 to 360). I believe that reporting this with averaged gain by model scaling and adding more verifications and allocating more descriptions of author’s findings can make this paper much more stronger.

I will increase the score once current concerns are addressed.

**Questions:**

- How do RL-trained models perform on this benchmark? Are there any inductive bias observed from visual encoders? Are there any underlying biases or trends from the training set? These points are revealed that these two points matter [2,3].
- Authors analyzed the order of video and image matters. Did the ordering of text inputs change performance?
- I believe that analysis on attention score can somehow verify the finding in Section 4.3 (line 400). How are the attention scores distributed towards image tokens?
- How can the evaluated models have forgetting issues (line 401) in the input sequence? Different from LSTM models that sequentially feed tokens to update hidden state, Transformers architecture processes the input tokens in parallel rather than sequentially. Isn’t the “forgetting” happens only when the input tokens are truncated according to the size of the context window? The size of the context window should be reported to make this claim concrete. In other words, how many previous tokens can the models condition on.

References

[1] Fang et al. MMBench-Video: A long-form multi-shot benchmark for holistic video understanding. NeurIPS D&B Track 2024.

[2] Wang et al. VideoHallucer: Evaluating Intrinsic and Extrinsic Hallucinations in Large Video-Language Models. arxiv:2406.16338

[3] Li et al. Vidhalluc: Evaluating Temporal Hallucinations in Multimodal Large Language Models for Video Understanding. CVPR 2025.

---

> ### Author Response · Authors · 2025-11-24
>
> > ### W1: Findings Are Merely Unsurprising Scaling Effects
>
> Thank you for this valuable feedback. We appreciate the opportunity to clarify how our contributions extend beyond standard scaling observations. While we acknowledge that model scale does play a role, our work reveals several important insights that go beyond the observation that "larger models perform better."
>
> **1. Establishing a novel and challenging benchmark.** Our primary contribution is the first benchmark specifically designed for image-grounded video perception and reasoning. The substantial performance gap—even the strongest model (Qwen2.5-VL-72B) achieves only 28.9% accuracy versus 88.8% for humans (Table 2)—demonstrates that current MLLMs are far from solving this task. This large gap reveals fundamental limitations in current models' ability to integrate external visual grounding with video understanding, representing an important and underexplored research direction.
>
> **2. Revealing nuanced patterns beyond standard scaling laws.** Our analysis uncovers several interesting patterns that provide insights beyond capacity scaling:
>
> - **Divergent scaling on perception vs. reasoning tasks (L368–376).** We observe that increasing model size yields asymmetric gains across task types. In the Qwen2.5-VL series, scaling from 7B to 72B provides substantial improvements on perception-oriented tasks (e.g., Existence, NLI) but more modest gains on reasoning-intensive tasks (e.g., Counting, Temporal Reasoning). This divergence suggests that current scaling approaches may disproportionately benefit perception capabilities while reasoning abilities present additional challenges—an observation with potentially important implications for model development priorities.
>
> - **Non-monotonic effects of multimodal grounding (Fig. 3(a)).** We observe interesting patterns in how models integrate multiple modalities. For smaller models, adding the image query can degrade performance compared to text-only queries. Furthermore, placing the image after video frames consistently outperforms placing it before for larger models, revealing positional bias and information forgetting effects in long multimodal sequences—nuanced behaviors that warrant further investigation.
>
> - **Visual token allocation trade-offs (Figs. 3(b), 4(a), 4(b)).** Our analysis reveals several practical insights: (i) increasing frame count consistently helps across models, (ii) resolution gains saturate beyond moderate levels, and (iii) under fixed token budgets, different model sizes favor different frame-resolution trade-offs. These findings offer practical guidance—suggesting to prioritize more frames over higher resolution under token constraints.
>
> **3. Task-specific model performance patterns.** Our analysis across 13 task categories (Table 3) reveals that different models excel at different capabilities, with no single architecture dominating across all tasks. These patterns may offer useful insights for model selection and future architecture design.
>
> We hope these findings can provide useful insights for future MLLM development in image-grounded video understanding, complementing the broader understanding of scaling effects in multimodal models.
>
> > ### W2: Lack of Experiments on Meta-data (e.g., Subtitles)
>
> We do not include subtitle-based experiments for two reasons. First, the videos we collected do not contain subtitles. Second, our benchmark is intentionally designed to evaluate models' ability to reason purely from the external image and visual video content, without relying on additional signals such as audio or subtitles. The presence or absence of subtitles does not affect the core image grounded video perception and reasoning capabilities that IV-Bench aims to measure.
>
> That said, we acknowledge that incorporating meta-data such as subtitles or audio cues could provide complementary information and represents a valuable direction for future extensions of this benchmark.

---

> ### Author Response · Authors · 2025-11-24
>
> > ### W3: All Tasks Use Multiple-Choice Questions
>
> We intentionally adopt a multiple-choice format because it offers several practical advantages while still effectively evaluating reasoning capabilities. Many well-established multimodal benchmarks—such as MMMU [1], VideoMME [2], and LongVideoBench [3] —rely on multiple-choice questions due to their clear evaluation protocol, automatic scoring, and high reproducibility. This format enables objective, consistent, and efficient evaluation at scale.
>
> In contrast, open-ended responses require extensive human annotation, introduce subjective variability in scoring, and significantly increase evaluation costs.
>
> [1] Yue X, Ni Y, Zhang K, et al. Mmmu: A massive multi-discipline multimodal understanding and reasoning benchmark for expert agi[C]//Proceedings of the IEEE/CVF Conference on Computer Vision and Pattern Recognition. 2024: 9556-9567.
>
> [2] Fu C, Dai Y, Luo Y, et al. Video-mme: The first-ever comprehensive evaluation benchmark of multi-modal llms in video analysis[C]//Proceedings of the Computer Vision and Pattern Recognition Conference. 2025: 24108-24118.
>
> [3] Wu H, Li D, Chen B, et al. Longvideobench: A benchmark for long-context interleaved video-language understanding[J]. Advances in Neural Information Processing Systems, 2024, 37: 28828-28857.
>
> > ### W4: Inter-annotator Agreement and Quality Control Rejection Rates
>
> Thank you for highlighting this important aspect. We appreciate the opportunity to provide detailed statistics on our annotation quality control process. To ensure the reliability of IV-Bench, we implement a rigorous "1-Generation, 3-Verification" pipeline, where each sample is generated by one annotator and independently verified by three other expert annotators.
>
> **Quality Control Process and Rejection Rates:**
>
> Our quality control process consists of two main rounds (see Appendix 6 for more details) with specific rejection rates at each stage:
>
> - **Initial Pool:** We begin with 1,500 videos and 4,500 questions (3 questions per video on average).
>
> - **First Round Quality Control:** This round focuses on the structure and content standardization of evaluation questions. We verify the clarity, precision, and unambiguity of each question, checking whether the query and options are clearly described and whether the answer is correct. We also confirm that the correct answers and distractors are both plausible and contextually relevant, ensuring each query can be accurately answered based on the provided video and image. Furthermore, we check task categorization accuracy, correcting any misclassifications to maintain consistency across all 13 predefined tasks. Samples with unfixable errors are removed. This round filters out 285 videos and 1,072 questions, resulting in 1,215 videos and 3,428 questions (retention rate: 81.0% for videos, 76.2% for questions).
>
> - **Second Round Quality Control:** Since some questions can be answered using only common sense or video content, we conduct a second round of quality control to eliminate leakage and ensure image necessity. During this phase, any query that can be resolved without the reference image or video is removed, and any text query that inadvertently reveals visual content is rewritten to eliminate leakage. We also pinpoint ineffective distractors—those easily dismissed using video alone—and manually introduce at least two effective distractors per question. These distractors are crafted so that, although incorrect for the current image, they would serve as the correct answers to alternative questions sharing the same text query but paired with a different image, thereby ensuring that the image is necessary for each sample. This stringent process further filters out 249 videos and 868 questions, yielding our final benchmark of 966 videos and 2,560 questions (overall retention rate: 64.4% for videos, 56.9% for questions).
>
> **Inter-Annotator Agreement:**
>
> To measure annotation consistency, we compute Fleiss' Kappa across all verification decisions made by the three independent inspectors. We achieve a Fleiss' Kappa of κ = 0.86, indicating strong inter-annotator agreement on quality standards, including whether distractors are effective, whether image information is necessary, and whether potential leakage exists.
>
> **Consensus Mechanism:**
>
> We apply a strict unanimous consensus rule: only samples where all three inspectors agree on validity (after any necessary rewriting or distractor augmentation) are retained in the final benchmark. Any unresolvable disagreement among inspectors results in sample removal. This conservative approach ensures high data quality at the cost of a lower retention rate, which we believe is justified given the importance of benchmark reliability.

---

> ### Author Response · Authors · 2025-11-24
>
> > ### W5: Model Scaling and Task-Specific Analysis
>
> Thank you for this constructive suggestion. We agree that analyzing performance gains by model scaling provides valuable insights into current MLLM capabilities. We have conducted a detailed analysis of scaling effects on **Perception Tasks** (e.g., Existence, Detailed Events) versus **Reasoning Tasks** (e.g., Counting, Temporal Reasoning) across multiple model families, including the latest Qwen2.5/3-VL and InternVL2.5/3 series.
>
> **1. Asymmetric Gains from Scaling:**
>
> We observe a consistent pattern where increasing model size yields significantly larger improvements on perception tasks compared to reasoning tasks. The table below summarizes the performance across major model families:
>
> | Model | Overall | Perception | Reasoning |
> |-------|---------|------------|-----------|
> | **Qwen2.5-VL-7B** | 18.5% | 18.9% | 17.9% |
> | **Qwen2.5-VL-32B** | 23.0% | 24.9% | 20.2% |
> | **Qwen2.5-VL-72B** | 28.9% | 33.7% | 21.9% |
> | *Gain (7B→72B)* | *+10.4%* | *+14.8%* | *+4.0%* |
> | | | | |
> | **Qwen3-VL-8B** | 18.2% | 20.1% | 15.4% |
> | **Qwen3-VL-32B** | 23.5% | 27.7% | 17.4% |
> | *Gain (8B→32B)* | *+5.3%* | *+7.6%* | *+2.0%* |
> | | | | |
> | **InternVL2.5-8B** | 17.4% | 17.8% | 16.5% |
> | **InternVL2.5-26B** | 20.6% | 22.4% | 18.1% |
> | **InternVL2.5-38B** | 26.6% | 30.4% | 21.1% |
> | **InternVL2.5-78B** | 28.6% | 33.4% | 21.9% |
> | *Gain (8B→78B)* | *+11.2%* | *+15.6%* | *+5.4%* |
> | | | | |
> | **InternVL3-8B** | 18.9% | 20.0% | 17.3% |
> | **InternVL3-14B** | 25.3% | 27.2% | 22.5% |
> | **InternVL3-38B** | 25.6% | 27.2% | 23.4% |
> | *Gain (8B→38B)* | *+6.7%* | *+7.2%* | *+6.1%* |
> | | | | |
> | **InternVL3.5-8B** | 19.3% | 18.9% | 19.8% |
> | **InternVL3.5-38B** | 26.0% | 27.3% | 24.0% |
> | *Gain (8B→38B)* | *+6.7%* | *+8.4%* | *+4.2%* |
>
> **2. Analysis of Findings:**
>
> *   **Perception Dominates Scaling Gains:** Across almost all model families, perception tasks see massive improvements as model size increases (e.g., Qwen2.5-VL: **+14.8%**, InternVL2.5: **+15.6%**). This confirms that larger parameters significantly enhance the model's ability to "see" and identify static content.
> *   **Reasoning Lags Behind:** In sharp contrast, reasoning tasks show much more modest gains (e.g., Qwen2.5-VL: **+4.0%**, InternVL2.5: **+5.4%**, Qwen3-VL: **+2.0%**, InternVL3.5: **+4.2%**). Even the largest 70B+ models struggle to break past the 22-23% accuracy mark on reasoning tasks, despite their strong perception capabilities (33%).
> *   **Implication:** This finding highlights a critical limitation of current scaling laws: while they effectively improve perception, they encounter severe diminishing returns for reasoning.

---

> ### Author Response · Authors · 2025-11-24
>
> > ### Q1: RL-Trained Models, Visual Encoder Biases, and Training Set Biases
>
> Thank you for raising these important questions about potential factors affecting model performance.
>
> **1. RL-trained models performance.** We have evaluated several models trained with reinforcement learning or preference optimization. The Qwen2.5-VL series (7B/32B/72B) incorporates RL training in their pipeline, achieving competitive performance (23.0%, 27.9%, and 28.9% respectively in Table 2). To further investigate the specific impact of RL/preference optimization, we conduct additional experiments comparing InternVL3.5-8B (base model) with InternVL3.5-8B-MPO (trained with Mixed Preference Optimization). The results show minimal difference: 19.2% for the base model versus 19.4% for the MPO variant (only +0.2%). This suggests that RL or preference optimization alone does not substantially improve performance on image-grounded video perception and reasoning, likely because such training paradigms primarily optimize for human preference alignment rather than the specific multimodal grounding and reasoning capabilities required by IV-Bench.
>
> **2. Inductive biases from visual encoders.** Isolating the effect of visual encoders is challenging because different MLLMs employ varying architectures, training data, and design choices simultaneously. For a controlled comparison, we would need models that differ only in their visual encoders while keeping all other components (language model, connector, training data, etc.) identical. Unfortunately, to our best knowledge, such carefully controlled model variants are not available in the current MLLM landscape. Most architectural differences are confounded with other factors—for example, InternVL models use InternViT while Qwen2.5-VL uses a different vision encoder, but they also differ in language backbones, training recipes, and data mixtures. Therefore, we cannot definitively attribute performance differences to visual encoder inductive biases alone. This represents an interesting direction for future controlled studies.
>
> **3. Training Set Biases and Data Distribution.** We acknowledge that the discrepancy between training data distribution and our benchmark's task formulation is a significant factor contributing to the low performance.
> *   **Data Formats:** Most current video-language models are trained on datasets consisting primarily of `(Video, Text)` pairs (e.g., captioning, standard QA) or `(Image, Text)` pairs.
> *   **Missing Modality Interaction:** The specific setting of **image-grounded video perception and reasoning**—where an external static image serves as a necessary grounding signal for video content—is extremely rare in current large-scale pre-training corpora.
> *   **OOD Generalization:** Consequently, IV-Bench effectively tests models in an **out-of-distribution (OOD)** setting. The consistent performance gap across all models (max 28.9% vs human 88.8%) strongly suggests that current models lack the learned representations to effectively bridge the semantic gap between an external reference image and a dynamic video without explicit training on such tri-modal data `(Video, Image, Text)`. This underscores the need for future datasets to include such grounding examples to bridge this capability gap.

---

> ### Author Response · Authors · 2025-11-24
>
> > ### Q2: Text Ordering Effects on Performance
>
> Thank you for this important question. As our task focuses on image-grounded video perception and reasoning, we primarily follow the widely adopted training settings used by most multimodal language models, where the textual query is placed after the visual information. This design choice aligns with common practices in recent MLLMs such as InternVL2.5 [1], LLaVA-OneVision [2], and Qwen2.5-VL [3], where the typical input order is: vision information → question.
>
> However, to thoroughly investigate whether text ordering affects model performance, we conduct additional experiments placing the text query before all visual content (i.e., question → video frames → image). The results reveal substantial performance degradation across all tested models:
>
> | Model | Text After Vision | Text Before Vision | Performance Drop |
> |-------|------------------:|-------------------:|-----------------:|
> | Minicpm-v | 17.2% | 13.4% | -3.8% |
> | Minicpm-o | 17.1% | 14.1% | -3.0% |
> | InternVL2.5-8B | 17.4% | 7.7% | -9.7% |
> | InternVL2.5-26B | 20.6% | 9.8% | -10.8% |
> | InternVL2.5-38B | 26.6% | 15.9% | -10.7% |
> | InternVL2.5-78B | 28.6% | 16.5% | -12.1% |
>
> **Key Findings:**
>
> **1. Text ordering significantly impacts performance.** Placing text before vision leads to substantial accuracy drops ranging from 3.0% to 12.1% across all models. This demonstrates that the relative positioning of textual and visual modalities plays a critical role in multimodal reasoning performance.
>
> **2. Different model families show varying sensitivity to text ordering.** The InternVL2.5 series exhibits particularly high sensitivity (9.7-12.1% drops), with larger models showing even greater degradation. In contrast, the MiniCPM series demonstrates more robustness (3.0-3.8% drops), suggesting that certain architectural designs or training strategies may better handle positional variations.
>
> **3. The substantial performance drops reveal critical insights into input ordering effects.** We hypothesize several possible explanations for the observed degradation. First and most importantly, the text-before-vision order likely deviates from the training configuration of these models. Most recent MLLMs are trained with vision-then-text ordering, and deviation from this training paradigm can significantly harm performance due to distribution shift. This hypothesis is strongly supported by the particularly severe drops in the InternVL2.5 series (up to 12.1%), suggesting these models are highly sensitive to input order mismatches. Second, these findings align with our forgetting and positional bias observations in Figure 3(a). When text appears before the lengthy visual sequence (video frames + image), the visual tokens that follow may dominate the attention distribution, causing the model to underutilize the earlier textual information despite all tokens being present in context. This attention dilution effect in long multimodal sequences demonstrates that input ordering consistently affects multimodal integration across different experimental settings.
>
> These findings confirm that text ordering does indeed affect performance, and our choice to follow the standard "vision-then-text" ordering is not only aligned with common practice but is also empirically optimal for image-grounded video understanding tasks.
>
> [1] Chen Z, Wang W, Cao Y, et al. Expanding performance boundaries of open-source multimodal models with model, data, and test-time scaling[J]. arXiv preprint arXiv:2412.05271, 2024.
> [2] Li B, Zhang Y, Guo D, et al. Llava-onevision: Easy visual task transfer[J]. arXiv preprint arXiv:2408.03326, 2024.
> [3] Bai S, Chen K, Liu X, et al. Qwen2. 5-vl technical report[J]. arXiv preprint arXiv:2502.13923, 2025.

---

> ### Author Response · Authors · 2025-11-24
>
> > ### Q3: Attention Score Analysis
>
> Thank you for this insightful suggestion. To quantitatively verify our findings regarding the impact of input ordering (Section 4.3), we analyzed the attention distribution in **InternVL3-8B** and **InternVL3-14B** models. We compared two configurations: placing the image *before* the video frames vs. placing the image *after* the video frames (closer to the text query).
>
> **Methodology:**
> We calculated the attention weights by aggregating the attention scores from all attention heads across all transformer layers during the decoding phase. Specifically, for each generated token, we computed the average attention weight it assigned to the visual tokens (image and video) across all layers and heads, and then averaged these values over the entire IV-Bench.
>
> The results are summarized below:
>
> | Model | Configuration | Accuracy | Image Attention | Video Attention |
> | :--- | :--- | :--- | :--- | :--- |
> | **InternVL3-8B** | Image Before Video | 16.19% | 0.0051 | 0.0865 |
> | **InternVL3-8B** | Image After Video | 20.21% | 0.0217 | 0.0730 |
> | **InternVL3-14B** | Image Before Video | 21.73% | 0.0041 | 0.0937 |
> | **InternVL3-14B** | Image After Video | 24.60% | 0.0218 | 0.0781 |
>
> **Analysis:**
>
> 1.  **Increased Attention on Image Tokens:** When the image is placed *after* the video (closer to the final prediction token), the average attention weight on image tokens increases drastically (approximately **4x** to **5x** higher) compared to the "Image Before Video" configuration (e.g., rising from 0.0051 to 0.0217 for InternVL3-8B).
> 2.  **Correlation with Performance:** This increase in image attention strongly correlates with the improvement in accuracy (from 16.19% to 20.21% for 8B, and 21.73% to 24.60% for 14B).
> 3.  **Low Visual Attention Phenomenon:** We also observe that the absolute attention scores allocated to visual tokens are generally low (e.g., ~0.02 for image, ~0.08 for video). This aligns with the findings in **FastV [1]**, which revealed that visual tokens often receive sparse attention in deeper layers of Large Vision-Language Models. Our results confirm this sparsity but highlight that even small shifts in this limited attention budget (from 0.005 to 0.02) can drive significant performance gains.
> 4.  **Verification of Positional Bias:** These attention statistics empirically verify the "forgetting" or positional bias effect discussed in Section 4.3. When the image appears early in the context window ("Before"), it receives significantly less attention, likely due to the dilution effect from the subsequent long video sequence. Placing the image later ("After") ensures it remains in the model's active focus, facilitating better visual grounding and improved reasoning performance.
>
> **Reference:**
> [1] Chen L, Zhao H, Liu T, et al. An image is worth 1/2 tokens after layer 2: Plug-and-play inference acceleration for large vision-language models[C]//European Conference on Computer Vision. Cham: Springer Nature Switzerland, 2024: 19-35.
>
> > ### Q4: Forgetting Issues in Transformer Models
>
> We appreciate the reviewer's question and clarify that our use of the term "forgetting" does not refer to truncation caused by exceeding the context window. All evaluated models operate well within their effective context limits. For example, Qwen2.5-VL has a context window of 128k tokens. In this model, every 28×28 pixel patch produces one visual token after projection. In our setting, a typical 420×240 frame produces approximately (420/28) × (240/28) ≈ 15 × 9 = 135 visual tokens. With 32 video frames + 1 query image, this yields about 33 × 135 ≈ 4,455 visual tokens, and the accompanying text (instruction, question, options) remains below 1,000 tokens. Thus, the total sequence length is approximately 5,000-6,000 tokens, which is well below the max token length in Qwen2.5-VL (128k).
>
> The "forgetting" we describe instead refers to a positional bias and attention dilution effect within the Transformer architecture. While you are correct that Transformers process tokens in parallel rather than sequentially like LSTMs, recent work has shown that even within the context window, earlier tokens in long multimodal sequences receive progressively less effective attention as sequence length grows. When the query image appears early in the sequence (before video frames), later video tokens tend to dominate the attention distribution, causing the model to under-utilize the earlier visual information despite all tokens being present in context. This phenomenon has been documented in recent multimodal language model research, where positional encoding and attention patterns can create implicit biases favoring more recent tokens in long sequences. This is why placing the query image after the video frames consistently improves performance—it ensures the critical grounding information appears in a position where it receives stronger attention.

---

> > ### Comment · Reviewer_2qaq · 2025-11-26
> >
> > I thank the authors for the detailed clarification and the additional experiments. The concerns regarding inter-annotator agreement have been addressed, and the response has effectively clarified the terminology regarding the 'forgetting' issues. Furthermore, the new analysis on attention scores and text ordering effects corroborates the findings and sound more complete.
> >
> > As a follow-up, do the authors have any hypotheses or proposed strategies for further enhancing model performance on this task?

---

> > > ### Author Response · Authors · 2025-11-26
> > >
> > > We sincerely appreciate your positive feedback. IV-Bench exposes a critical gap: current MLLMs often fail to effectively utilize external visual cues, achieving only 28.9% accuracy compared to 88.8% for humans. Although the primary focus of this work is to establish the benchmark, based on our analysis (Table 2), we propose a concrete roadmap for future improvements across three dimensions:
> > >
> > > **1. Data-Centric Strategy: Bridging the Gap and Enhancing Reasoning**
> > >
> > > * **Constructing IV-Triplets:** Current pre-training data rarely contains `(Video, External Image, Text)` triplets. We plan to construct synthetic datasets where static objects are explicitly paired with video segments to move the task from Out-Of-Distribution (OOD) to in-distribution.
> > > * **Augmenting Underperforming Tasks:** Our results (Table 2) show that models perform significantly worse on specific tasks like **Counting** and **Temporal Reasoning** compared to general perception. Future data construction must prioritize these "weak links" by incorporating video samples heavily annotated with count-based and timestamp-based instruction data to explicitly strengthen these capabilities.
> > >
> > > **2. Model Architecture: Injecting Temporal Dynamics**
> > >
> > > * **Addressing the "Bag-of-Frames" Limitation:** The poor performance on temporal tasks (e.g., Temporal Reasoning, Space-Time Computing) indicates that current models, which largely rely on static image encoders, suffer from a "Bag-of-Frames" limitation [1,2]—processing video as a collection of independent images without true temporal awareness. To counter this, future architectures should integrate **time-aware modules** (e.g., temporal adapters) directly into the visual encoder. This would allow the model to capture the dynamic evolution of visual content and motion dependencies *before* the features enter the LLM, significantly improving the representation of dynamic content.
> > >
> > > **3. Training Strategy: Curriculum Learning**
> > >
> > > * **Perceive-then-Reason Curriculum:** We propose a **Curriculum Learning** strategy to handle the complexity of IV tasks. Models should first be trained on easier **Perception tasks** (e.g., Existence, Appearance) to establish strong visual grounding. Subsequently, they can be progressively trained on complex **Reasoning tasks** (e.g., Counting, Temporal Ordering). This step-by-step approach prevents the model from being overwhelmed by difficult reasoning requirements before establishing basic perception capabilities.
> > >
> > > **Future Roadmap:**
> > > Building on these hypotheses, our immediate future plan is to leverage **IV-Bench**, along with other recent multi-video benchmarks such as Mvbench, TempCompass[3], MVU-Eval[4], as a guidance. We aim to train a robust MLLM specifically optimized for complex temporal and image-grounded video understanding, verifying the effectiveness of the strategies proposed above.
> > >
> > > [1] Li K, Wang Y, He Y, et al. Mvbench: A comprehensive multi-modal video understanding benchmark[C]//Proceedings of the IEEE/CVF Conference on Computer Vision and Pattern Recognition. 2024: 22195-22206.
> > >
> > > [2] Buch S, Eyzaguirre C, Gaidon A, et al. Revisiting the" video" in video-language understanding[C]//Proceedings of the IEEE/CVF conference on computer vision and pattern recognition. 2022: 2917-2927.
> > >
> > > [3] Liu Y, Li S, Liu Y, et al. Tempcompass: Do video llms really understand videos?[J]. arXiv preprint arXiv:2403.00476, 2024.
> > >
> > > [4] Peng T, Wang H, Zhang Y, et al. MVU-Eval: Towards Multi-Video Understanding Evaluation for Multimodal LLMs. NIPS2025.

---

> > > > ### Comment · Reviewer_2qaq · 2025-11-26
> > > >
> > > > Thank you for suggesting future work on data, architecture and objective-centric approaches. All of the approaches sound promising. I believe that including all the content we discussed in the main script would make the paper more concrete. I raised the score to 6.

---

### Official Review · Reviewer_MRHL · 2025-10-31

**Soundness:** 3
**Presentation:** 3
**Contribution:** 3
**Rating:** 6
**Confidence:** 4

**Summary:**

This paper introduces IV-Bench, the first comprehensive benchmark designed specifically for evaluating Multimodal Large Language Models (MLLMs) on image-grounded video perception and reasoning tasks. The benchmark comprises 966 videos and 2,560 meticulously annotated image-text queries, spanning 13 distinct tasks (7 perception and 6 reasoning) across 5 video categories. The authors conduct an extensive evaluation of 28 state-of-the-art MLLMs, revealing that even the best-performing model (Qwen2.5-VL-72B) achieves only 28.9% overall accuracy, which is significantly lower than human performance (88.8%). Furthermore, through ablation studies, the authors analyze the impact of factors such as the order of image input, number of video frames, and resolution on model performance, providing valuable insights for future model design.

**Strengths:**

1. IV-Bench is the first benchmark specifically dedicated to image-grounded video understanding tasks. It effectively addresses the limitation of existing video benchmarks that rely solely on text queries, thereby advancing the field of multimodal reasoning evaluation.
2.The benchmark's high quality and validity are ensured through a two-round quality control process, the use of externally sourced images, and diverse task design. The inclusion of "effective distractors" is particularly noteworthy, as it forces models to rely on the image information.
3. The systematic evaluation of 28 models thoroughly exposes the significant shortcomings of current MLLMs on this task. The ablation studies offer practical design recommendations, such as the finding that placing the image after the video frames yields better performance.

**Weaknesses:**

1.Although covering multiple categories, the total of 966 videos is smaller than some existing video understanding benchmarks (e.g., Video-Bench with 5,917 videos), which might affect the benchmark's broad representativeness.
2.The current work focuses only on the triplet input of image-text-video. It does not explore more complex multimodal grounding signals, such as audio, multiple images, or dynamic image sequences (e.g., GIFs).
3.While emphasizing "image-grounding," the benchmark may not fully account for whether a single static image is sufficient to represent dynamic changes in a video. Some tasks might still lean towards static matching rather than genuine spatio-temporal reasoning.

**Questions:**

1.	During the construction of IV-Bench, did you consider incorporating dynamic image sequences (e.g., GIFs or short video clips) as the grounding signal to better simulate real-world scenarios where users provide visual context?
2.	Regarding the particularly poor performance on tasks like "Temporal Reasoning," have you conducted further analysis into the root causes? Is it related to the models' ability to understand long videos or their inherent temporal modeling mechanisms?

---

> ### Author Response · Authors · 2025-11-24
>
> > ### W1: Benchmark Scale and Representativeness
>
> We appreciate this concern about benchmark scale. However, we believe the comparison should consider the specific benchmark category and the number of evaluation samples rather than just video count. As shown in Table 1, among benchmarks with text+image queries (the same setting as IV-Bench), IV-Bench contains 966 videos and 2,560 questions, which is substantially larger than comparable benchmarks: VideoRefer-BenchQ (198 videos, 1,000 questions) and V2P-Bench (980 videos, 1,172 questions). **Our question count is 2.2× larger than the second-largest benchmark** in this category, providing sufficient samples for stable and reliable model evaluation.
>
> Furthermore, while some text-only video benchmarks contain more videos, the evaluation cost increases significantly with video count. Our current scale **balances comprehensive coverage with practical evaluation efficiency**, making it feasible for the community to evaluate models while maintaining representativeness across 13 diverse task categories and 9 source domains.
>
> > ### W2 & Q1: Limited to Image-Text-Video Triplet
>
> We appreciate this suggestion for future extensions. IV-Bench focuses on the image-text-video triplet because this formulation is **both common and practical in many real-world scenarios**, such as e-commerce product retrieval, media content analysis, and video editing assistance. In these applications, users typically provide a single static image as the reference signal, which proves sufficient for most grounding tasks.
>
> From a research perspective, **IV-Bench establishes a fundamental framework for image-grounded video perception and reasoning**. We fully agree that incorporating richer modalities—such as audio, multiple reference images, or dynamic image sequences—would broaden the scope and capabilities. Our benchmark naturally serves as a foundation that can be extended to these additional modalities in future work.
>
> > ### W3: Static Matching vs. Spatio-Temporal Reasoning
>
> We appreciate this concern and would like to clarify the role of images in our benchmark. The spatio-temporal reasoning we evaluate **does not depend on the external image itself being dynamic**. Rather, the image primarily serves to ground the target entity in the video and to supplement the text query with essential visual cues that text alone cannot fully specify. The question design is what reveals the model's ability to perform genuine spatio-temporal reasoning over video content.
>
> IV-Bench is a comprehensive benchmark that includes diverse task types. While some tasks involve static matching, many others require strong spatio-temporal reasoning capabilities. For example, in our Space-Time Computing task, a video may feature multiple hosts, while the query image shows only one of them. A representative question asks: "How long does the person in the image speak during his/her first appearance?" To answer this correctly, the model must: (1) first ground the specific person in the video using the external image, and (2) then conduct temporal reasoning over the corresponding video segments to calculate the speaking duration. This process goes far beyond static matching and directly evaluates the model's spatio-temporal reasoning capability.

---

> ### Author Response · Authors · 2025-11-24
>
> > ### Q2: Root Cause Analysis of Poor Temporal Reasoning Performance
>
> Thank you for this insightful question regarding the root causes of poor performance on Temporal Reasoning tasks. To investigate whether this limitation stems from insufficient visual information (sampling limitations) or inherent model capabilities, we conduct a controlled experiment using Qwen3-VL-8B with increasing frame counts (32, 64, and 128 frames) without decreasing the resolution.
>
> **Experimental Results: Information vs. Capability**
>
> Increasing frames from 32 to 128 substantially improves temporal information availability, as shown in the table below. We define **Temporal Coverage** as the percentage of samples where at least one frame falls within the ground truth (GT) relevant time segment, and **Frame Density** as the average number of frames sampled per second.
>
> | Metric | 32 Frames | 128 Frames | Improvement |
> |--------|-----------|------------|-------------|
> | **Temporal Coverage** | 76.09% | 85.94% | +9.85 pp (+13.0%) |
> | **Frame Density** | 0.13 fps | 0.41 fps | +3.2× (+216%) |
>
> We analyze four frame-sensitive tasks with similarly low baseline performance (<20%) to isolate the bottleneck:
>
> | Task | 32 Frames | 128 Frames | Change |
> |------|-----------|------------|--------|
> | **Counting** | 14.98% | 16.48% | +1.50% |
> | **Space-Time Computing** | 14.02% | 16.36% | +2.34% |
> | **Keyframe Extraction** | 18.47% | 17.77% | -0.70% |
> | **Temporal Reasoning** | 13.33% | 11.11% | **-2.22%** |
>
> **Key Findings:**
>
> 1.  **Divergent Response to More Information:** While information-aggregation tasks (Counting, Space-Time Computing) benefit from more frames, Temporal Reasoning performance actually **declines**.
> 2.  **The Paradox:** If the bottleneck were simply insufficient information, Temporal Reasoning should improve alongside other tasks. The decline suggests that additional frames introduce **information overload** rather than providing useful signals for this specific task.
>
> **Root Cause Analysis:**
>
> This result strongly suggests that the poor performance is due to **inherent deficiencies in temporal modeling mechanisms**, specifically the **"Static Bias"** identified in recent studies like MVBench [1]. Current MLLMs heavily rely on static image encoders (e.g., CLIP) trained to align single images with text. This design causes models to process videos by **"averaging" static frame content** rather than modeling the temporal flow, leading to the loss of dynamic inter-frame information. Consequently, models exhibit a significant **"Temporal-Spatial Performance Gap,"** where increasing frame density fails to improve (or even degrades) performance on temporal reasoning tasks due to the lack of genuine causal modeling.
>
> [1] Li K, Wang Y, He Y, et al. Mvbench: A comprehensive multi-modal video understanding benchmark[C]//Proceedings of the IEEE/CVF Conference on Computer Vision and Pattern Recognition. 2024: 22195-22206.
>
> **Conclusion:**
>
> The bottleneck for Temporal Reasoning is the model's inherent capability to model temporal relationships, order, and causality. Without explicit temporal reasoning mechanisms, simply increasing frame count fails to improve performance and may even degrade it due to context noise.

---

### Official Review · Reviewer_AHVG · 2025-10-31

**Soundness:** 3
**Presentation:** 3
**Contribution:** 4
**Rating:** 8
**Confidence:** 4

**Summary:**

IV-Bench is one of the first benchmark for image-grounded video perception and reasoning in MLLMs, featuring 966 videos and 2,560 external image-text queries across 13 diverse tasks. Unlike other benchmarks, all queries use external images as anchors, increasing real-world complexity. Evaluations on 28 leading MLLMs show performance is much lower than humans, especially in reasoning. The paper analyzes model size, reasoning methods, and token allocation, and both dataset and code are open-source.

**Strengths:**

- This is one of the first systematic benchmark proposal for image-grounded video perception and reasoning, filling a notable gap in existing evaluation ecosystems.

- The benchmark features high data quality and a robust evaluation framework. All images are sourced externally, preventing information leakage and accurately reflecting the difficulty of cross-source visual grounding in complex tasks such as search and retrieval, which strongly enhances the validity of the assessment.

- Experimental analyses are thorough and the findings are insightful for follow-up research. The paper details the effects of image token order, scale, resolution, and frame count on model performance, and clearly points out the extremely limited reasoning capabilities of current MLLMs in image-grounded video scenarios. It also shows model scaling has minimal impact on reasoning, which offers clear direction for future advances.

**Weaknesses:**

- Some tasks (e.g., Instruction Understanding, Summary) feature only weak connections to the visual “grounding,” where images act in a mainly auxiliary role and do not fully embody the core definition of image grounding.

- The paper lacks granular error analysis. It reports only accuracy without dissecting which sub-tasks exhibit systematic failure (e.g., universal shortcomings in temporal reasoning or attribute change), and does not offer failure case examples. It is suggested to add error type breakdowns (e.g., missed visual cues, text comprehension errors, temporal confusion) and provide typical failed cases versus human labels in the appendix.

- The discussion around the semantic gap and selection logic for external images should be strengthened. Although the necessity and realism of using external images is emphasized, there is no structured description or case analysis of their diversity/difficulty. A quantitative contrast between tasks with highly similar external images (near frame extraction) versus high-gap scenarios should be considered.

**Questions:**

1. On subjective scoring consistency and ground-truth diversity: For reasoning tasks with subjective or diverse answers, how is human label consistency and answer distribution measured? How are multi-answer questions evaluated?

2. On the irreplaceability of images: Has image replacement testing been conducted across all task categories? If an image is replaced by another with similar semantics but different visuals, does the answer change? If not, how can it be demonstrated that the image truly “grounds” the reasoning process?

3. On prompt strategy: In Table 2, do all models use the optimal prompt order (image after video)? If not, have you considered rerunning the experiments for fairness?

---

> ### Author Response · Authors · 2025-11-24
>
> > ### W1: Weak Visual Grounding in Some Tasks
>
> We understand the concern, but this is not true in IV-Bench. When creating IV-Bench, we deliberately designed and filtered samples so that the image is required to determine the correct answer. For example, the tasks like Summary and Instruction Understanding, videos typically contain multiple people or objects, while the query image corresponds to only one specific person or object. Different entities in the same video yield different correct answers, so the model must first use the visual information in the image to ground the corresponding entity in the video and then answer based on its role or behavior. This visual grounding process is essential for answering the question, where the image plays a necessary role rather than an auxiliary one.
>
> > ### W2: Lack of Granular Error Analysis
>
> Thank you for this valuable suggestion. Following your feedback, we have conducted some granular error analysis to identify the failure patterns in mllms. The analysis reveals:
>
> **Frame Sensitivity and Temporal Reasoning Analysis:** We hypothesize that performance on tasks requiring fine-grained understanding (e.g., Temporal Reasoning, Counting, Space-Time Computing) might be limited by sparse frame sampling (typically 8-32 frames), which misses critical temporal details. To investigate whether this limitation stems from insufficient visual information or inherent model capabilities, we conduct a controlled experiment using Qwen3-VL-8B with increasing frame counts (32 vs 128 frames).
>
> **Experimental Results: Information vs. Capability**
>
> Increasing frames from 32 to 128 substantially improves temporal information availability, as shown in the table below. We define **Temporal Coverage** as the percentage of samples where at least one frame falls within the ground truth (GT) relevant time segment, and **Frame Density** as the average number of frames sampled per second.
>
> | Metric | 32 Frames | 128 Frames | Improvement |
> |--------|-----------|------------|-------------|
> | **Temporal Coverage** | 76.09% | 85.94% | +9.85 pp (+13.0%) |
> | **Frame Density** | 0.13 fps | 0.41 fps | +3.2× (+216%) |
>
> We analyze four frame-sensitive tasks with similarly low baseline performance (<20%) to isolate the bottleneck:
>
> | Task | 32 Frames | 128 Frames | Change |
> |------|-----------|------------|--------|
> | **Counting** | 14.98% | 16.48% | +1.50% |
> | **Space-Time Computing** | 14.02% | 16.36% | +2.34% |
> | **Keyframe Extraction** | 18.47% | 17.77% | -0.70% |
> | **Temporal Reasoning** | 13.33% | 11.11% | **-2.22%** |
>
> **Key Findings:**
>
> 1.  **Divergent Response to More Information:** While information-aggregation tasks (Counting, Space-Time Computing) benefit from more frames, Temporal Reasoning performance actually **declines**.
> 2.  **The Paradox:** If the bottleneck were simply insufficient information, Temporal Reasoning should improve alongside other tasks. The decline suggests that additional frames introduce **information overload** rather than providing useful signals for this specific task.
>
> **Analysis:**
>
> This result strongly suggests that the poor performance is due to **inherent deficiencies in temporal modeling mechanisms**, specifically the **"Static Bias"** identified in recent studies like MVBench [1]. Current MLLMs heavily rely on static image encoders (e.g., CLIP) trained to align single images with text. This design causes models to process videos by **"averaging" static frame content** rather than modeling the temporal flow, leading to the loss of dynamic inter-frame information. Consequently, models exhibit a significant **"Temporal-Spatial Performance Gap,"** where increasing frame density fails to improve (or even degrades) performance on temporal reasoning tasks due to the lack of genuine causal modeling.
>
> **Relationship Explanation (RE) struggles:** Our experiments reveals that models perform particularly poorly on RE tasks when dealing with negative descriptions or absent relationships. This finding aligns with recent work showing that vision-language models struggle with negation understanding (Vision-Language Models Do Not Understand Negation [1]). Our benchmark provides a evaluation of this limitation in the context of image-grounded video perception and reasoning.
>
> We will incorporate those error analysis into the revised manuscript. Additionally, we will provide detailed error type breakdowns (e.g., missed visual cues, text comprehension errors, temporal confusion) and typical failed cases versus human labels in the Appendix.
>
> [1] Alhamoud K, Alshammari S, Tian Y, et al. Vision-language models do not understand negation[C]//Proceedings of the Computer Vision and Pattern Recognition Conference. 2025: 29612-29622.

---

> ### Author Response · Authors · 2025-11-24
>
> > ### W3: Necessity of External Images and Semantic Gap Analysis
>
> Thank you for this valuable suggestion. We conduct a comprehensive quantitative analysis to demonstrate the necessity and structured design of using external images with diverse semantic gaps.
>
> **Experimental Setup:**
> We quantify the semantic gap by calculating the cosine similarity between the external image and 8 uniformly sampled video frames using CLIP-ViT-L/14. Based on the similarity scores of 2,560 samples, we divide the dataset into three groups:
> - **High-Similarity (Top 25%, >0.567):** Scenarios akin to near-frame extraction.
> - **Mid-Similarity (Mid 50%, 0.427-0.567):** Moderate semantic gap.
> - **Low-Similarity (Bottom 25%, <0.427):** High semantic gap scenarios.
>
> We evaluate 10 mllms across these groups. The results reveal critical insights that justify our design:
>
> **1. Diverse Semantic Gaps are Essential for Comprehensive Evaluation:**
> Different models exhibit distinct strengths depending on the semantic gap. We evaluate 10 state-of-the-art models and present their performance across similarity groups in the table below:
>
> | Model | High-Sim | Mid-Sim | Low-Sim | Diff (H-L) | Preference |
> |-------|----------|---------|---------|------------|------------|
> | **Aria-32B** | 19.59% | 15.22% | **19.91%** | -0.31% | Balanced |
> | **Qwen2.5-VL-7B** | **20.85%** | 16.71% | 19.28% | +1.57% | Near-Frame |
> | **Qwen3-VL-8B** | 19.28% | 16.71% | **20.22%** | -0.94% | Balanced |
> | **Qwen3-VL-32B** | 20.53% | 23.29% | **27.12%** | -6.58% | Semantic Gap |
> | **InternVL2.5-26B** | **22.10%** | 19.29% | 21.94% | +0.16% | Balanced |
> | **InternVL3-8B** | **21.94%** | 16.63% | 20.69% | +1.25% | Near-Frame |
> | **InternVL3-14B** | 25.39% | 24.47% | **27.12%** | -1.72% | Semantic Gap |
> | **InternVL3-38B** | 26.49% | 24.47% | **27.43%** | -0.94% | Balanced |
> | **InternVL3.5-8B** | **21.79%** | 17.80% | 19.75% | +2.04% | Near-Frame |
> | **InternVL3.5-8B-MPO** | **22.26%** | 18.35% | 18.97% | +3.29% | Near-Frame |
>
> The results reveal divergent model behaviors: some models (e.g., Qwen3-VL-32B) excel in high-gap scenarios (+6.58%), while others (e.g., InternVL3.5-8B-MPO) prefer near-frame scenarios (+3.29%). This underscores that a holistic assessment must include diverse semantic gaps to evaluate both fine-grained visual correspondence and semantic reasoning capabilities.
>
> **2. Task-Specific Dependencies and Discriminative Power:**
> We analyze performance across our 13 task categories and observe distinct patterns. The table below summarizes the average performance across all 11 models for each task type:
>
> | Task Type | High-Sim | Mid-Sim | Low-Sim | Diff (H-L) | Dependency |
> |-----------|----------|---------|---------|------------|------------|
> | **Detailed Events** | **35.37%** | 21.44% | 14.73% | +20.64% | Strong High-Sim |
> | **Summary** | **26.38%** | 18.28% | 16.39% | +9.99% | High-Sim |
> | **Instruction Understanding** | 17.96% | **21.86%** | 11.62% | +6.34% | High-Sim |
> | **Existence** | **20.00%** | 15.70% | 16.18% | +3.82% | High-Sim |
> | **Reverse Existence** | **16.51%** | 14.70% | 13.39% | +3.12% | High-Sim |
> | **Keyframe Extraction** | **16.81%** | 14.05% | 13.93% | +2.88% | High-Sim |
> | **Counting** | **20.45%** | 18.11% | 19.66% | +0.80% | Neutral |
> | **Space-Time Computing** | **18.53%** | 12.95% | 18.18% | +0.35% | Neutral |
> | **Temporal Reasoning** | 12.03% | 11.55% | **12.50%** | -0.47% | Neutral |
> | **Spatial Relationship** | 32.66% | 25.52% | **34.85%** | -2.19% | Low-Sim |
> | **Constrained OCR** | 31.49% | 29.55% | **36.64%** | -5.16% | Low-Sim |
> | **Natural Language Inference** | 28.57% | 28.89% | **36.36%** | -7.79% | Strong Low-Sim |
> | **Attribute Change** | 18.02% | 22.48% | **28.48%** | -10.47% | Strong Low-Sim |
>
> The impact of the semantic gap varies by task nature. Tasks relying on direct visual correspondence (e.g., **Detailed Events**, **Existence**) benefit significantly from high visual similarity, with **Detailed Events** showing a +20.6% advantage in near-frame settings. Conversely, tasks requiring semantic abstraction or differentiation (e.g., **Attribute Change**, **Natural Language Inference**) perform better with larger semantic gaps. Notably, **Attribute Change** achieves 10.5% higher accuracy in low-similarity settings, suggesting that high-similarity images may introduce bias by fixing attention on static states, whereas low-similarity images encourage focus on attribute transformations.
>
> Furthermore, low-similarity samples demonstrate **39.7% higher standard deviation** in model performance compared to high-similarity samples. This indicates that scenarios with larger semantic gaps provide stronger differentiation between model capabilities.

---

> ### Author Response · Authors · 2025-11-24
>
> > ### Q1: Subjective Scoring Consistency and Answer Diversity
>
> We appreciate this concern about evaluation objectivity. IV-Bench is designed exclusively with multiple-choice questions where each question has a single correct answer, following the established evaluation protocol of widely adopted benchmarks such as VideoMME [1] and MMMU [2]. This design choice offers several key advantages: (1) evaluation is straightforward and objective without requiring human annotation for scoring, (2) it avoids the substantial human labor costs and potential inconsistencies associated with subjective scoring of open-ended responses, and (3) it ensures reproducible evaluation across different models and settings.
>
> [1] Fu C, Dai Y, Luo Y, et al. Video-mme: The first-ever comprehensive evaluation benchmark of multi-modal llms in video analysis[C]//Proceedings of the Computer Vision and Pattern Recognition Conference. 2025: 24108-24118.
>
> [2] Yue X, Ni Y, Zhang K, et al. Mmmu: A massive multi-discipline multimodal understanding and reasoning benchmark for expert agi[C]//Proceedings of the IEEE/CVF Conference on Computer Vision and Pattern Recognition. 2024: 9556-9567.
>
> > ### Q2: Image Irreplaceability
>
> We appreciate this insightful question and have conducted explicit tests to verify the irreplaceability of images. As shown in Appendix (L1250–1274), we perform an image-caption replacement experiment using Qwen2.5-VL-32B and Qwen2.5-VL-72B. For each query image, we first use the model to generate a detailed caption, then replace the image with this caption and feed video + caption to the model. If the image were not essential, the caption—containing similar high-level semantics—should yield performance comparable to video + image. However, the results reveal a substantial gap: for Qwen2.5-VL-32B, video-only achieves 18.0%, video + caption reaches 20.9% (+2.9%), while video + image achieves 23.0% (+5.0%). For Qwen2.5-VL-72B, the pattern is similar: video-only at 18.6%, video + caption at 24.8% (+6.2%), and video + image at 28.9% (+10.3%). The significantly larger performance gain from adding raw images compared to semantic captions demonstrates that visual details in the image, not just textual semantics, are necessary to answer the queries. Therefore, the image is indeed irreplaceable in IV-Bench.
>
> > ### Q3: Prompt Strategy Consistency
>
> Yes, we ensure fairness across all models. As stated in Sec. 4.1 (Line 321, page 6), all models evaluated in Table 2 use the same default input order: "video frames + image + question". This corresponds to the optimal prompt strategy identified in our analysis, ensuring that all models are evaluated under consistent and favorable conditions.

---

### Author Response · Authors · 2025-11-29

We thank the reviewers for their constructive feedback and are encouraged by their **unanimous positive assessment** of IV-Bench.

### 1. Recognition of Strengths
The reviewers unanimously recognize IV-Bench as a pioneering and timely contribution that fills a critical gap in the multimodal evaluation ecosystem. They specifically commend:

* **Rigorous Data Construction:** The reviewers **praise** our **commitment to** **pure manual annotation** rather than automated pipelines, ensuring high data quality. The design of **"effective distractors"** and the use of external images to prevent information leakage are highlighted as robust measures to ensure valid assessment (Reviewer MRHL, 2qaq).
* **Comprehensive Evaluation & Insights:** The extensive benchmarking of **28 state-of-the-art MLLMs** **is recognized** as impressive and diagnostic. Reviewers note that our analysis effectively exposes the significant performance gap between models and humans, while our ablation studies (e.g., on input ordering and resolution) offer **actionable guidance** for future model design (Reviewer Zjuc, MRHL).

### 2. Enhancements during Rebuttal
To address specific concerns and further strengthen the manuscript, we conduct extensive additional analyses:

* **Deep-Dive Error Analysis:** We perform controlled experiments on frame scaling and attention mechanisms, identifying "Static Bias" and "input ordering effects" as key factors (addressing Reviewer MRHL & Zjuc).
* **Quality Assurance Metrics:** We provide detailed statistics on our quality control pipeline, including a Fleiss’ Kappa of 0.86, validating the reliability of our annotations (addressing Reviewer 2qaq).
* **Experimental Validation:** We add experiments regarding text-input ordering, image irreplaceability, and attention score distribution to empirically support our findings.
* **Future Roadmap:** We outline concrete directions for future work, including data-centric construction, architectural improvements for temporal modeling, and curriculum learning strategies.

### 3. Conclusion
Following these clarifications and additional experiments, **Reviewer 2qaq increases their score from 4 to 6.**, and **Reviewers MRHL and Zjuc affirm their positive ratings**. We are confident that IV-Bench serves as a valuable resource for the community.

---

### Meta-Review · Area_Chair_Unsw · 2026-01-07

**Summary:**

The reviewers unanimously recognize IV-Bench as a pioneering and high-quality benchmark that fills a critical void in Multimodal Large Language Model (MLLM) evaluation: Image-Grounded Video Perception and Reasoning. Unlike traditional video benchmarks that rely on text prompts, IV-Bench requires models to use an external static image to ground entities and reason over temporal dynamics. The benchmark comprises 966 videos and 2,560 meticulously annotated queries across 13 tasks. Extensive evaluation of 28 SOTA models (e.g., GPT-4o, InternVL2.5) reveals a massive performance gap compared to humans (28.9% vs. 88.8%).

**Reviewer Concerns:**

Most of the concerns were addressed by the rebuttal. And the only negative score (4) before rebuttal was raised to 6.

**Reviewer Scores:**

The final scores should be around 8/6/6/6.

---

### Decision · Program_Chairs · 2026-01-26

Accept (Poster)